# OPENVID-1M: A LARGE-SCALE HIGH-QUALITY DATASET FOR TEXT-TO-VIDEO GENERATION

**Kepan Nan**[1*] **Rui Xie**[1*] **Penghao Zhou**[2*] **Tiehan Fan**[1]
**Zhenheng Yang**[2] **Zhijie Chen**[2] **Xiang Li**[3] **Jian Yang**[1] **Ying Tai**[1†]
[1] State Key Laboratory for Novel Software Technology, Nanjing University
[2] ByteDance    [3] Nankai University

## ABSTRACT

Text-to-video (T2V) generation has recently garnered significant attention thanks to the large multi-modality model Sora. However, T2V generation still faces two important challenges: 1) Lacking a precise open sourced high-quality dataset. The previously popular video datasets, *e.g.*WebVid-10M and Panda-70M, overly emphasized large scale, resulting in the inclusion of many low-quality videos and short, imprecise captions. Therefore, it is challenging but crucial to collect a precise high-quality dataset while maintaining a scale of millions for T2V generation. 2) Ignoring to fully utilize textual information. Recent T2V methods have focused on vision transformers, using a simple cross attention module for video generation, which falls short of making full use of semantic information from text tokens. To address these issues, we introduce `OpenVid-1M`, a precise high-quality dataset with expressive captions. This open-scenario dataset contains over 1 million text-video pairs, facilitating research on T2V generation. Furthermore, we curate 433K 1080p videos from `OpenVid-1M` to create `OpenVidHD-0.4M`, advancing high-definition video generation. Additionally, we propose a novel Multi-modal Video Diffusion Transformer (MVDiT) capable of mining both structure information from visual tokens and semantic information from text tokens. Extensive experiments and ablation studies verify the superiority of `OpenVid-1M` over previous datasets and the effectiveness of our MVDiT. Project webpage is available at https://nju-pcalab.github.io/projects/openvid.

## 1 INTRODUCTION

Text-to-video (T2V) generation, which aims to create a video sequence based on the condition of a text describing the video, is an emerging visual understanding task. Thanks to the significant advancements of large multi-modality model Sora (Brooks et al., 2024), T2V generation has recently garnered significant attention. For example, based on DiT (Peebles & Xie, 2023), OpenSora[1], OpenSoraPlan (Lab & etc., 2024) and recent works (Wang et al., 2023c; Lu et al., 2023) utilize the collected million-scale text-video datasets to reproduce Sora. However, these diffusion models (Ma et al., 2024a; Wang et al., 2023a; Lu et al., 2023; Chen et al., 2023c; Wang et al., 2023c) still face two critical challenges: 1) *Lacking a precise high-quality video dataset*. Previously popular video datasets, such as WebVid-10M and Panda-70M, overly emphasized large scale, resulting in the inclusion of many low-quality videos and short, imprecise captions. Therefore, collecting a precise high-quality text-to-video dataset while maintaining a scale of millions is challenging but crucial for T2V generation. 2) *Ignoring to fully utilize textual information*. Recent T2V methods have focused on vision transformer (*e.g.*, STDiT in OpenSora), using a simple cross attention module, which falls short of making full use of semantic information from text tokens.

In this work, we curate a precise high-quality dataset named `OpenVid-1M`, which comprises over 1 million in-the-wild video clips, all with resolutions of at least $512 \times 512$, accompanied by detailed captions. As shown in Figure 1, our `OpenVid-1M` has several characteristics: 1) *Superior in quantity*: Compared to specific-scenario datasets like UCF-101, which are typically tailored for *particular contexts with limited video clips*, our `OpenVid-1M` stands out as a million-level dataset designed for open scenarios,enhancing model generalization and enabling video generation across diverse scenes.

---

* Equal contributions. † Corresponding author: yingtai@nju.edu.cn.

[1]https://github.com/hpcaitech/Open-Sora

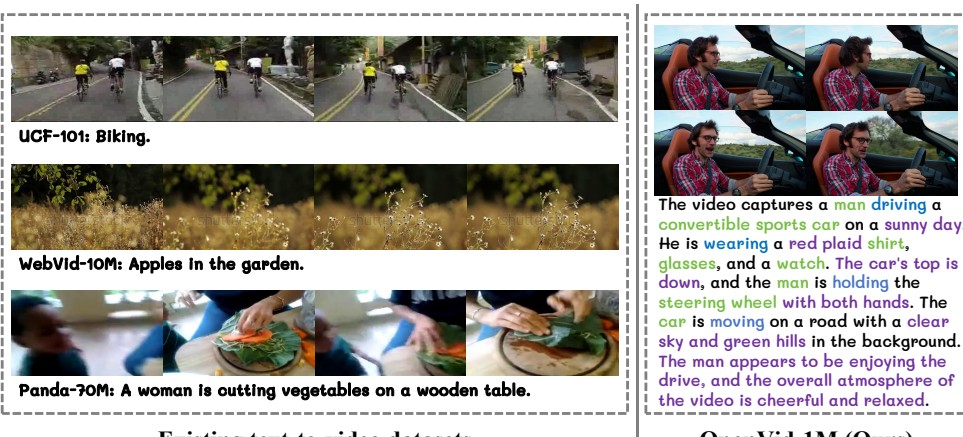

**Existing text-to-video datasets** | **OpenVid-1M (Ours)**

Figure 1: **Comparison of OpenVid-1M to the existing text-to-video datasets.** Specific-scenario datasets like UCF-101 contain low rasolution videos with simple captions (categories), WebVid-10M contains low-quality videos with watermarks and Panda-70M contains many flickering (or still) and blurry videos along with imprecise captions. In contrast, our OpenVid-1M contains a million high-quality video clips coupled with expressive and precise captions (we highlight nouns in green, verbs in blue, and easily overlooked details in purple).

2) *Superior in visual quality*: OpenVid-1M is strictly selected from the aspects of aesthetics, temporal consistency, motion difference, and clarity assessment. We also curate OpenVidHD-0.4M to advance research in high-definition video generation. Specifically, OpenVid-1M far outstrips the commonly-used WebVid-10M (Bain et al., 2021) in both *resolution and video quality*, as WebVid-10M includes low-quality, watermarked videos. Meanwhile, Panda-70M (Chen et al., 2024b) contains many videos with *low aesthetics, static, flickering, excessively dynamic or poor clarity*, whereas our OpenVid-1M is curated to ensure high-quality visuals across various aspects. 3) *Expressive in caption*: Specific-scenario datasets like UCF-101 use category labels as captions, while datasets like WebVid-10M and Panda-70M often have short, imprecise captions. In contrast, our OpenVid-1M provides expressive captions, enabling the generation of rich, coherent video content through the multimodal model LLaVA-v1.6-34b (Liu et al., 2024).

To address the second challenge, we propose a novel Multi-modal Video Diffusion Transformer (MVDiT). Unlike previous DiT architectures (Lab & etc., 2024; Ma et al., 2024a) that focus on modeling the visual content, our MVDiT features a parallel visual-text architecture to *mine both structure information from visual tokens and semantic information from text tokens* to improve the video quality. MVDiT extracts visual and text tokens, combines them into a multi-modal feature, and enhances token interaction through a self-attention module. Then, a multi-modal temporal-attention module ensures semantic and structural consistency, while a multi-head cross-attention module integrates text semantics into visuals.

Our contributions are threefold: 1) We introduce OpenVid-1M, a million-level high-quality dataset with expressive captions for facilitating video generation. 2) We validate OpenVid-1M on T2V task with two models, *i.e.* STDiT and our proposed MVDiT, which makes full use of semantic information from text tokens to improve visual quality. 3) We further demonstrate the superiority of OpenVid-1M on video restoration task.

## 2 RELATED WORK

**Text-to-video Datasets.** Existing text-to-video training datasets can be categorized into two classes: Specific-scenario and open-scenario. Specific-scenario datasets (Yu et al., 2023a; Yuan et al., 2024; Rossler et al., 2019; Soomro et al., 2012; Xiong et al., 2018; Siarohin et al., 2019) typically consist of a limited number of text-video pairs collected for specific contexts. For example, UCF-101 (Soomro et al., 2012) is a action recognition dataset which contains 101 classes and 13,320 videos in total. Taichi-HD (Siarohin et al., 2019) contains 2,668 videos recording a single person performing Taichi. ChronaMagic (Yuan et al., 2024) comprises 2,265 high-quality time-lapse videos with accompanying

Table 1: Data processing pipeline. The first three steps can be processed in parallel to enhance processing efficiency, while the subsequent steps are processed sequentially.

| Pipeline | Tool | Computation Resources | Processing Time (hours) | Remark |
|---|---|---|---|---|
| Aesthetics score | LAION Aesthetics Predictor | 32 A100 | 320 | Get high aesthetics score set $S_A$ |
| Temporal consistency | CLIP (Radford et al., 2021) | 48 A100 | 173 | Obtain moderate consistency set $S_T$ |
| Motion difference | UniMatch (Xu et al., 2023) | 48 A100 | 59 | Obtain moderate amplitude of motion Set $S_M$ |
| Intersection of qualified videos | Intersection | - | - | Obtain intersection: $S_I = S_A \cap S_T \cap S_M$ |
| Clarity assessment | DOVER-Technical (Wu et al., 2023) | 8 A100 | 25 | Obtain clear and high-quality video set $S$ |
| Clip extraction | Cascaded Cut Detector (Blattmann et al., 2023) | 8 A100 | 30 | Split multi-scene videos: $\widetilde{S} = Detector(S)$ |
| Video caption | LLaVA-v1.6-34b (Liu et al., 2023a) | 8 A100 | 46 | Obtain long captions for the videos |

text descriptions. As a pioneering open-scenario T2V dataset, WebVid-10M (Bain et al., 2021) comprises 10.7 million text-video pairs with a total of 52K video hours. Panda-70M (Chen et al., 2024b) collects 70 million high-resolution and semantically coherent video samples. Recently, InternVid (Wang et al., 2023d) proposes a scalable approach for autonomously constructing a video-text dataset using large language models, resulting in 234 million video clips with text descriptions. However, WebVid-10M contains low-quality videos with watermarks, Panda-70M contains lots of static, flickering, low-clarity videos along with short captions, while InternVid primarily focuses on video understanding tasks. In contrast, our `OpenVid-1M` comprises over 1 million high-quality in-the-wild video clips, with 433K in 1080p resolution, accompanied by expressive captions.

**Text-to-video Models.** Current text-to-video generation methods can be divided into: UNet (Khachatryan et al., 2023; Ge et al., 2023; Wang et al., 2023a; Chen et al., 2023a; 2024a; Yu et al., 2023b; Zeng et al., 2023), and DiT based methods (Ma et al., 2024a; Chen et al., 2023c; Lu et al., 2023). UNet based methods have been widely studied. Modelscope (Wang et al., 2023a) introduces a spatio-temporal block and a multi-frame training strategy to enhance text-to-video synthesis, achieving State-of-the-Art (SOTA) results. VideoCrafter (Chen et al., 2024a) investigates the feasibility of leveraging low-quality videos and synthesized high-quality images to obtain a high-quality video model. DiT has demonstrated significant superiority and efficiency in image generation task (Chen et al., 2023b; 2024c; Zhao et al., 2024) and has recently attracted considerable attention in the field of video generation. Sora (Brooks et al., 2024) revolutionizes video generation. Latte (Ma et al., 2024a) employs a simple and general video Transformer as the backbone to generate videos. Recently, OpenSora[1], trained based on a pretrained T2I model (Chen et al., 2023b) and a large text-to-video dataset, aims to reproduce Sora. In contrast to the previous DiT structures, we propose a novel MVDiT that features a parallel visual-text architecture, mining both structure information from visual tokens and semantic information from text tokens to improve the video quality.

## 3 CURATING `OpenVid-1M`

This section outlines the date processing steps in Table 1. `OpenVid-1M` is curated from ChronoMagic, CelebvHQ (Zhu et al., 2022), Open-Sora-plan (Lab & etc., 2024) and Panda[2]. Since Panda is much larger than the others, here we primarily describe the filtering details on our downloaded Panda-50M.

**Aesthetics Score.** Visual aesthetics are crucial for video content satisfaction and pleasure. To enhance text-to-video generation, we filter out videos with low aesthetics scores using the LAION Aesthetics Predictor. This results in a subset $S_A$ with the top 20% highest-scoring videos from Panda-50M. For the other three datasets, we select the top 90% to form subset $S'_A$.

**Temporal Consistency.** Video clips with temporal consistency are crucial for training. We use CLIP (Radford et al., 2021) to extract visual features and measure temporal consistency by analyzing cosine similarity between adjacent frames. Clips with high scores (nearly static) and low scores (frequent flickering) are filtered out, yielding a suitable subset, $S_T$, from Panda-50M. For the other datasets with good temporal consistency, no filtering is performed.

**Motion Difference.** We employ UniMatch (Xu et al., 2023) to assess optical flow as a motion difference score, selecting videos with smooth movement, since temporal consistency alone is insufficient to filter out high-speed objects that still maintain consistency. Videos with high flow scores, indicating rapid motion, are *unsuitable* for training. We filter out clips with the highest and

---

[2]Since we can only download 50M, we refer to this version Panda-50M in this work.

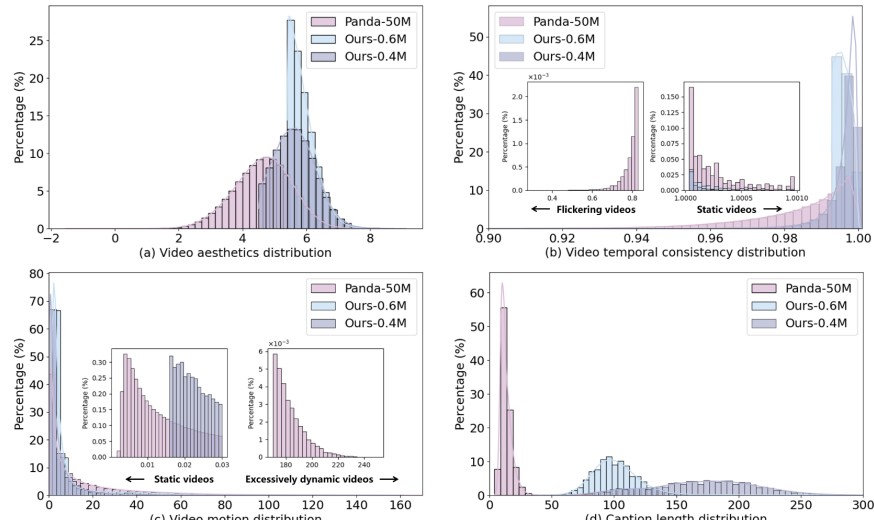

Figure 2: **Comparisons on video statistics** between `OpenVid-1M` and Panda-50M.

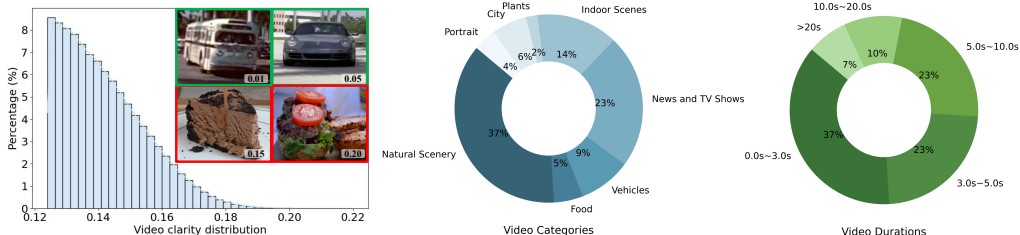

Figure 3: **Left:** Clarity distribution of `OpenVid-1M`. We also present 4 samples to visualize the clarity differences. Samples outlined in green contour are blurry with low clarity scores, while those outlined in red contour are clearer with high clarity. **Middle & Right:** `OpenVid-1M` contains diverse distributions of video category and duration.

lowest scores in Panda-50M to create subset $S_M$. For the other three datasets, we derive a subset, $S'_M$, without applying the remaining processing steps. Instead, we directly calculate the intersection of $S'_A$ and $S'_M$ to obtain $S' = S'_A \cap S'_M$, *i.e.,* Ours-0.4M illustrated in Figure 2.

**Clarity Assessment.** High-clarity videos are essential for T2V generation. Since Panda-50M contains many blurry clips, we filter those with very low clarity, as shown in Figure 3. We calculate the intersection of the three sets from Panda to obtain $S_I = S_A \cap S_T \cap S_M$, resulting in aesthetically pleasing, stable videos with smooth movement. Using the DOVER (Wu et al., 2023) model, we estimate the DOVER-Technical score for each clip in $S_I$ and retain high-clarity videos with clean textures. Finally, we select the top 30% of clips with the highest scores to form the video set $S$.

**Clip Extraction.** Beyond the aforementioned steps, some video clips may contain multiple scenes, thus we introduce the Cascaded Cut Detector (Blattmann et al., 2023) to split multi-scene clips in $S$ to achieve clip extraction, ensuring each contains only one scene. After clip extraction, we obtain $\widetilde{S}$ from Panda-50M, *i.e.,* Ours-0.6M illustrated in Figure 2.

**Video Caption.** As highlighted in Sora technical report, detailed captions greatly benefit video generation. After obtaining the video clip set, we recaption them using large multimodal model, LLaVA-v1.6-34b (Liu et al., 2023a), to create expressive descriptions. Since CelebvHQ lacks captions, we also provide captions for its video clips. Figure 2(d) compares the length of our prompts with those in Panda-50M, showing our expressive prompts offer a significant advantage by providing richer semantic information. We compile our high-quality dataset, `OpenVid-1M` (*i.e.,* Ours-0.6M + Ours-0.4M). Additionally, we meticulously select 1080p videos from `OpenVid-1M` to construct `OpenVidHD-0.4M`, *advancing the High-Definition (HD) video generation within the community*.

Table 2: Comparisons with previous text-to-video datasets. Our `OpenVid-1M` is a *million-level, high-quality and open-scenario* video dataset for training high-fidelity text-to-video models.

| Dataset | Scenario | Video clips | Average length (seconds) | Duration (hours) | Resolution | Caption |
|---|---|---|---|---|---|---|
| UCF101 | Action | 13K | 7.2 | 2.7 | 320×240 | N/A |
| Taichi-HD | Human | 3K | - | - | 256×256 | N/A |
| SkyTimelapse | Sky | 35K | - | - | 640×360 | N/A |
| FaceForensics++ | Face | 1K | - | - | Diverse | N/A |
| WebVid | Open | 10M | 18.7 | 52k | 596×336 | Short |
| InternVid | Open | 234M | 11.7 | 760.3K | Diverse | Short |
| ChronoMagic | Metamorphic | 2K | 11.4 | 7 | Diverse | Long |
| CelebvHQ | Portrait | 35K | 6.6 | 65 | 512×512 | N/A |
| OpenSoraPlan-V1.0 | Open | 400K | 24.5 | 274 | 512×512 | Long |
| Panda | Open | 70M | 8.5 | 166k | Diverse | Short |
| `OpenVid-1M` (Ours) | Open | 1M | 7.2 | 2.1k | Diverse | Long |
| `OpenVidHD-0.4M` (Ours) | Open | 433K | 9.6 | 1.2k | 1920×1080 | Long |

## 4 DATA PROCESSING AND STATISTICAL COMPARISON

**Data Processing Differences against SVD (Blattmann et al., 2023)**. Our data processing pipeline draws inspiration from the SVD pipeline, yet several distinctions exist: 1) *Visual quality evaluation*: Both SVD and our `OpenVid-1M` utilize an aesthetic predictor to retain highly aesthetic videos. Additionally, `OpenVid-1M` integrates the recent model DOVER (Wu et al., 2023) to assess the video clarity, preserving high-quality videos with clean textures. 2) *Motion evaluation*: SVD utilizes the traditional Farneback optical flow method and RAFT (Teed & Deng, 2020) to estimate optical flows. In contrast, `OpenVid-1M` adopts a more efficient UniMatch (Xu et al., 2023) to achieve better optical flows, addressing not only static videos but also those with fast movements. 3) *Time consistency evaluation*: SVD employs clip extraction solely to prevent sudden video changes, whereas `OpenVid-1M` additionally removes flicker videos. 4) *Processing efficiency*: SVD initially extracts video clips and then filters from a large pool, while `OpenVid-1M` first selects high-quality videos and then extracts clips, significantly enhancing processing efficiency. Finally, `OpenVid-1M` will be made publicly available, while the training dataset in SVD is not.

**Comparison with Panda-50M**. The statistical comparisons between `OpenVid-1M` and Panda-50M are illustrated in Figure 2. 1) *Video Aesthetics Distribution*: Our subsets Ours-0.6M and Ours-0.4M exhibit higher aesthetics scores compared to Panda-50M, suggesting superior visual quality. 2) *Video Motion Distribution*: Our subsets display a higher proportion of videos with moderate motion, implying smoother and more consistent motion. Conversely, Panda-50M appears to contain numerous videos with flickering and static scenes. 3) *Video Temporal Consistency Distribution*: Our subsets exhibit a more balanced distribution of moderate temporal consistency values, whereas Panda-50M includes videos with either static or excessively dynamic motion. 4) *Caption Length Distribution*: Our subsets feature significantly longer captions than Panda-50M, providing richer semantic information. Overall, `OpenVid-1M` demonstrates superior quality and descriptive richness, particularly in aesthetics, motion, temporal consistency, caption length and clarity as well.

**Comparisons with Other Text-to-video Datasets.** We compare our `OpenVid-1M` and `OpenVidHD-0.4M` to several previous datasets in Table 2. We also present video categories and durations statistics of `OpenVid-1M` in Figure 3. As shown, `OpenVid-1M` is a *million-scale, high-quality and open-scenario* video dataset designed for training high-fidelity text-to-video models. Specifically, `OpenVid-1M` consists of 1,019,957 clips, averaging 7.2 seconds each, with a total video length of 2,051 hours. Compared to previous million-level datasets, WebVid-10M contains low-quality videos with watermarks and Panda-70M contains many still, flickering, or blurry videos along with short captions. In contrast, our `OpenVid-1M` contains high-quality, clean videos with dense and expressive captions generated by the large multimodal model LLaVA-v1.6-34b. Additionally, compared to previous high-quality datasets that are usually designed for specific scenarios with limited video clips, our `OpenVid-1M` is a large-scale dataset for open scenarios, including portraits, scenic views, cityscapes, metamorphic content, *etc*.

## 5 METHOD

Inspired by MMDiT (Esser et al., 2024), we propose a Multi-modal Video Diffusion Transformer (MVDiT) architecture. Shown in Figure 4, its architecture diverges from prior methods (Lab & etc.,

2024; Ma et al., 2024a) by emphasizing a parallel visual-text structure for mining both structure from visual tokens and semantic from text tokens. Each MVDiT layer encompasses four steps: Initial extraction of visual and linguistic features, integration of a novel Multi-Modal Temporal-Attention module for improved temporal consistency, facilitation of interaction via Multi-Modal Self-Attention and Multi-Head Cross-Attention modules, and subsequent forwarding to the final feedforward layer.

## 5.1 FEATURE EXTRACTION

Given a video clip, we adopt a pre-trained variational autoencoder to encode input video clip into features in latent space. After being corrupted by noise, the obtained video latent is input into a 3D patch embedder to model the temporal information. Then, we add positional encodings and flatten patches of the noised video latent to a patch encoding sequence $\mathbf{X} \in \mathbb{R}^{T \times C \times HW}$. Following Chen et al. (2023b), We input the text prompts into the T5 large language model (Raffel et al., 2020) for conditional feature extraction. Then, we embed the text encoding to match the channel dimension of the visual tokens to obtain the text tokens $\hat{\mathbf{Y}} \in \mathbb{R}^{C \times L}$, where $L$ represents the length of the text tokens. Finally, we take the text and noised visual tokens as input of MVDiT.

## 5.2 MULTI-MODAL VIDEO DIFFUSION TRANSFORMER

**Multi-Modal Self-Attention Module.** We design a Multi-Modal Self-Attention (MMSA) module. Text tokens $\hat{\mathbf{Y}}$ are repeated by $T$ times along the temporal dimension to generate $\mathbf{Y} \in \mathbb{R}^{T \times C \times L}$. We adopt adaptive layer normalization both in text branch and visual branch to encode timestep information into the model. Then, we concatenate the visual tokens with text tokens to generate the multi-modal feature $\mathbf{F}^s \in \mathbb{R}^{T \times C \times (HW+L)}$, which is input into the MMSA module containing a Self-Attention Layer (SAL):

$$\mathbf{F}_{\text{SAL}}^s = \text{SAL}(\text{Concat}(\text{AdaLN}(\mathbf{X}, \mathbf{t}_1), \text{AdaLN}(\mathbf{Y}, \mathbf{t}_1))) \tag{1}$$

$$\text{AdaLN}(\mathbf{X}, \mathbf{t}_1)) = \gamma_1^1 \text{LayerNorm}(\mathbf{X}) + \beta_1^1. \tag{2}$$

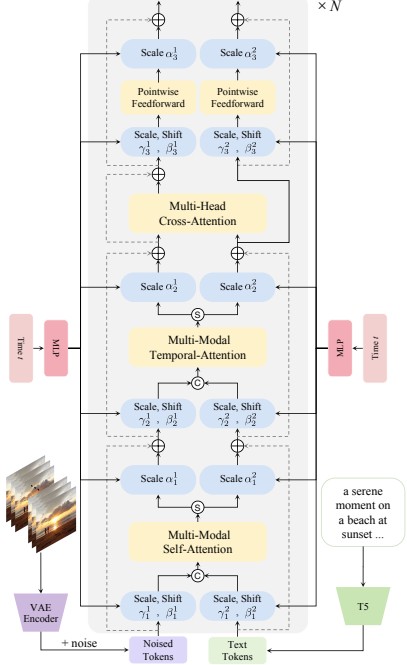

Figure 4: **Overview of MVDiT** with parallel visual-text architecture. Concatenation is indicated by ⓒ and split is indicated by ⓢ.

The self-attention operation is conducted to promote the interaction between visual tokens and text tokens in each frame, which can be implemented easily with matrix multiplication. Notably, since each video frame is paired with a unique text prompt, the text tokens vary across frames after the SAL, where they receive structural information from different frames. Then, we split the visual tokens and text tokens from the enhanced multi-modal features. Following Chen et al. (2023b), we also regress dimension-wise scaling parameter $\alpha$, which is applied before residual connections within the Transformer block. It can be formulated as follows:

$$\mathbf{X}_{\text{SAL}}^s, \mathbf{Y}_{\text{SAL}}^s = \text{Split}(\mathbf{F}_{\text{SAL}}^s), \ \mathbf{X}^s = \mathbf{X} + \alpha_1^1 \mathbf{X}_{\text{SAL}}^s, \ \mathbf{Y}^s = \mathbf{Y} + \alpha_1^2 \mathbf{Y}_{\text{SAL}}^s. \tag{3}$$

**Multi-Modal Temporal-Attention Module.** After obtaining the enhanced visual features and text features, we build a Multi-Modal Temporal-Attention (MMTA) module on the top of the MMSA to efficiently capture temporal information. Unlike the temporal attention used in previous methods (Lab & etc., 2024; Ma et al., 2024a), we consider capturing temporal information from both the text features and the visual features. Specifically, we concatenate the tokens from two branches to obtain the multi-modal feature $\mathbf{F}^t \in \mathbb{R}^{T \times C \times (HW+L)}$. We then input $\mathbf{F}^t$ into the MMTA module, where a Temporal-Attention Layer (TAL) is used to conduct communication along the temporal dimension:

$$\mathbf{X}_{\text{TAL}}^t, \mathbf{Y}_{\text{TAL}}^t = \text{Split}(\text{TAL}(\text{Concat}(\text{AdaLN}(\mathbf{X}^s, \mathbf{t}_2), \text{AdaLN}(\mathbf{Y}^s, \mathbf{t}_2)))), \tag{4}$$

$$\mathbf{X}^t = \mathbf{X}^s + \alpha_2^1 \mathbf{X}_{\text{SAL}}^t, \ \mathbf{Y}^t = \mathbf{Y}^s + \alpha_2^2 \mathbf{Y}_{\text{SAL}}^t. \tag{5}$$

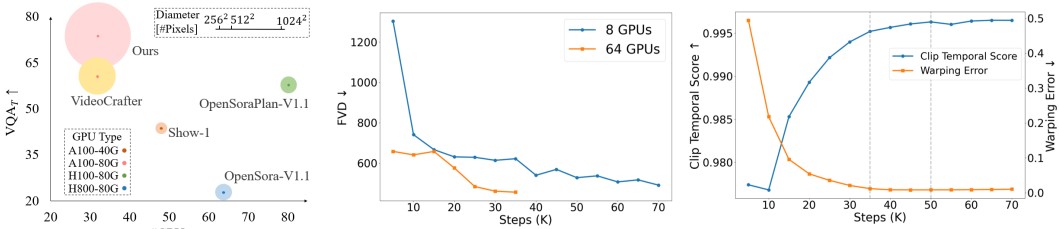

Figure 5: **Left:** Comparison with SOTA T2V models on VQA$_T$, GPU type and resolution. The color of the middle dot in each circle indicates GPU type, and circle diameter represents video resolution. **Middle:** Curves of FVD with different number of GPUs. More GPUs accelerates the model's convergence. **Right:** Curves of clip temporal score and warping error. Our T2V model typically starts to stabilize at 35K steps and achieves temporal consistency around 50K steps.

This design enables the model to learn both structural temporal consistency from visual information and semantic temporal consistency from textual information. For simplicity, temporal positional embedding is omitted.

**Multi-Head Cross-Attention Module.** While the MMSA module merges tokens from both modalities for attention, T2V still requires an explicit process to embed semantic information from text tokens into visual tokens. The absence of semantic information may impair video generation performance. Therefore, we developed a Multi-Head Cross-Attention (MHCA) module, in which a Cross-Attention Layer (CAL) is introduced to facilitate direct communication between text and visual tokens. Specifically, we take the flattened visual tokens $\mathbf{X}^t \in \mathbb{R}^{T \times C \times HW}$ as Query and text tokens $\mathbf{Y}^t \in \mathbb{R}^{T \times C \times L}$ as Key and Value, and input them into a cross-attention layer:

$$\mathbf{X}^c = \text{CAL}(\mathbf{X}^t, \mathbf{Y}^t) + \mathbf{X}^t. \tag{6}$$

Afterward, both the visual and text tokens are passed through a feedforward layer. Since a single MVDiT layer updates both token types, this process can be repeated iteratively to enhance video generation performance. After N iterations, the final visual feature is used to predict noise and covariance at time $t$. Our MVDiT is inspired by MMDiT, whose effectiveness has been thoroughly validated. Importantly, our work is the first to emphasize a parallel visual-text structure for extracting structural information from visual tokens and semantic information from text tokens in T2V generation.

# 6 EXPERIMENTS

## 6.1 EXPERIMENTAL SETTINGS

**Datasets and Evaluation Metrics.** We adopt proposed `OpenVid-1M` to train our MVDiT. `OpenVidHD-0.4M` is used further for HD video generation. WebVid-10M and Panda-50M are adopted for dataset comparisons. We evaluate our model on public benchmark in Liu et al. (2023b), which evaluates text-to-video generation model based on visual quality, text-video alignment and temporal consistency. Specifically, we adopt aesthetic score (VQA$_A$) and technical score(VQA$_T$) for video quality assessment. We evaluate the alignment of input text and generated video in two aspects, including image-video consistency (SD_score) and text-text consistency (Blip_bleu). We also evaluate temporal consistency of generated video with warping error and semantic consistency (Clip_temp_score).

**Implementation Details.** We use Adam (Kingma & Ba, 2014) as optimizer, and the learning rates is set to $2e-5$. We sample video clips containing 16 frames at 3-frame intervals in each iteration. We adopt random horizontal flips and random crop to augment the clips during the training stage. All experiments are conducted on NVIDIA A100 80G GPUs. We adopt PixArt-$\alpha$ (Chen et al., 2023b) for weight initialization and employ T5 model as the text encoder. The training process starts with 256×256 models, whose weights are then used to train 512×512 models, and these in turn serve as pretrained weights for 1024×1024 models. Starting with low-resolution training equips the model with coarse-grained modeling capabilities, while subsequent high-resolution finetuning enhances its ability to capture fine details. This staged approach reduces both computational cost and overall training time compared to directly starting with high-resolution training.

Table 3: Comparison with state-of-the-art text-to-video generation methods. The best results are marked in **bold**, while the second best ones are underscored.

| Method | Resolution | Training Data | VQA$_A$↑ | VQA$_T$↑ | Blip_bleu↑ | SD_score↑ | Clip_temp_score↑ | Warping_error↓ |
|---|---|---|---|---|---|---|---|---|
| Lavie (Wang et al., 2023c) | 512×320 | Vimeo25M | 63.77 | 42.59 | 22.38 | 68.18 | 99.57 | 0.0089 |
| Show-1 (Zhang et al., 2023) | 576×320 | WebVid-10M | 23.19 | 44.24 | 23.24 | 68.42 | 99.77 | 0.0067 |
| OpenSora-V1.1 | 512×512 | Self collected-10M | 22.04 | 23.62 | 23.60 | 67.66 | 99.66 | 0.0170 |
| Latte (Ma et al., 2024a) | 512×512 | Self collected-330K | 55.46 | 48.93 | 22.39 | 68.06 | 99.59 | 0.0203 |
| VideoCrafter (Chen et al., 2023a) | 1024×576 | WebVid-10M; Laion-600M | 66.18 | 58.93 | 22.17 | 68.73 | 99.78 | 0.0295 |
| Modelscope (Wang et al., 2023b) | 1280×720 | Self collected-Billions | 40.06 | 32.93 | 22.54 | 67.93 | 99.74 | 0.0162 |
| Pika | 1088×612 | Unknown | 59.09 | 64.96 | 21.14 | 68.57 | **99.97** | **0.0006** |
| OpenSoraPlan-V1.2 (Lab & etc., 2024) | 640×480 | Self collected-7.1M | 23.25 | 65.86 | 19.93 | **69.21** | **99.97** | 0.001 |
| CogVideoX-5B (Yang et al., 2024) | 720×480 | Self collected-35M | 35.12 | **76.86** | **24.21** | 68.91 | 99.79 | 0.0077 |
| Ours | 1024×1024 | OpenVid-1M | **73.46** | 68.58 | 23.45 | 68.04 | 99.87 | 0.0052 |

Table 4: Comparisons with previous representative text-to-video training datasets. The STDiT model used in OpenSora is adopted and kept the same for all of the cases. For fair comparison, training iterations are selected at the same step (50K) for fair comparison. All models with 256×256 resolution are adequately trained on 32 A100 GPUs for at least 14 days to reach 50K iterations.

| Resolution | Training Data | VQA$_A$↑ | VQA$_T$↑ | Blip_bleu↑ | SD_score↑ | Clip_temp_score↑ | Warping_error↓ |
|---|---|---|---|---|---|---|---|
| 256×256 | WebVid-10M (Bain et al., 2021) | 13.40 | 13.34 | 23.45 | 67.64 | 99.62 | 0.0138 |
| 256×256 | Panda-50M (Chen et al., 2024b) | 17.08 | 9.60 | 24.06 | 67.47 | 99.60 | 0.0200 |
| 256×256 | OpenVid-1M (Ours) | 17.78 | 12.98 | **24.93** | 67.77 | 99.75 | 0.0134 |
| 1024×1024 | WebVid-10M (4× Super-resolution) | 69.26 | 65.74 | 23.15 | 67.60 | 99.64 | 0.0137 |
| 1024×1024 | Panda-50M (4× Super-resolution) | 63.25 | 53.21 | 23.60 | 67.44 | 99.57 | 0.0163 |
| 1024×1024 | Panda-50M-HD | 13.48 | 42.89 | 21.78 | **68.43** | 99.84 | 0.0136 |
| 1024×1024 | OpenVidHD-0.4M (Ours) | **73.46** | **68.58** | 23.45 | 68.04 | **99.87** | **0.0052** |

## 6.2 COMPARISON WITH STATE-OF-THE-ART MODELS

In this section, we evaluate our method's performance and compare it with other models. For each model, we employ a consistent set of 700 prompts from Liu et al. (2023b) to generate videos. Metrics from Liu et al. (2023b) is used to evaluate the quality of generated videos.

**Quantitative Evaluation.** The comparison between our method and others is summarized in Table 3 and Figure 5. Our model achieves the highest VQA$_A$ (73.46%) and the second best VQA$_T$ (68.58%), indicating superior video aesthetics and clarity. Additionally, it achieves the second best Clip_temp_score (99.87%), demonstrating its good ability on temporal consistency. Overall, our model shows robust performance across various metrics while using less training data, demonstrating the superiority of our OpenVid-1M, highlighting its effectiveness in text-to-video generation tasks.

The comparison between OpenVid-1M and previous representative text-to-video training datasets is listed in Table 4. We adopt STDiT model used in OpenSora for all of the cases. In $256 \times 256$ resolution, the model trained with our OpenVid-1M generates the best scores across all metrics except VQA$_T$. This is reasonable, as the low resolution of the videos prevents showcasing the high quality of OpenVid-1M. The similar conclusion can be found in $1024 \times 1024$ resolution results, indicating the superiority of OpenVid-1M in generating high-quality videos. Moreover, our OpenVidHD-0.4M can be directly used to train high-definition (*e.g.,* $1024 \times 1024$) videos, whereas WebVid-10M cannot and Panda-50M has not yet undergone resolution- and quality-level filtering. To compare results at $1024 \times 1024$ resolution, we use ×4 video super-resolution to generate $1024 \times 1024$ videos from models trained on WebVid-10M and Panda-50M. Clearly, training with our OpenVidHD-0.4M yields better scores than combining other datasets with super-resolution. We also manually select 1080P videos from Panda-50M to create Panda-50M-HD for a fairer comparison. We can see that the model trained on our OpenVid-1M demonstrates superior performance, while the model trained on Panda-50M-HD performs poorly. This discrepancy may be attributed to the low-quality videos in Panda-50M-HD (*e.g.*, low aesthetics and clarity, nearly static scenes, and frequent flickering), a problem our data processing pipeline effectively avoids.

**Qualitative Evaluation.** Visual comparisons are shown in Figure 6. The first column demonstrates that our model generates clearer, more aesthetically pleasing and more detailed videos due to our high-resolution OpenVid-1M. In the second example, our model demonstrates a strong ability on prompt understanding, accurately depicting the 'android' and 'surrounded by colorful Easter eggs' from the text. In the third column, unrealistic dust appears in front of the car in videos from Lavie and VideoCrafter, while our model better captures the 'kicking up dust', highlighting its superior motion quality. We emphasize our method's ability to generate clearer and more aesthetically pleasing videos

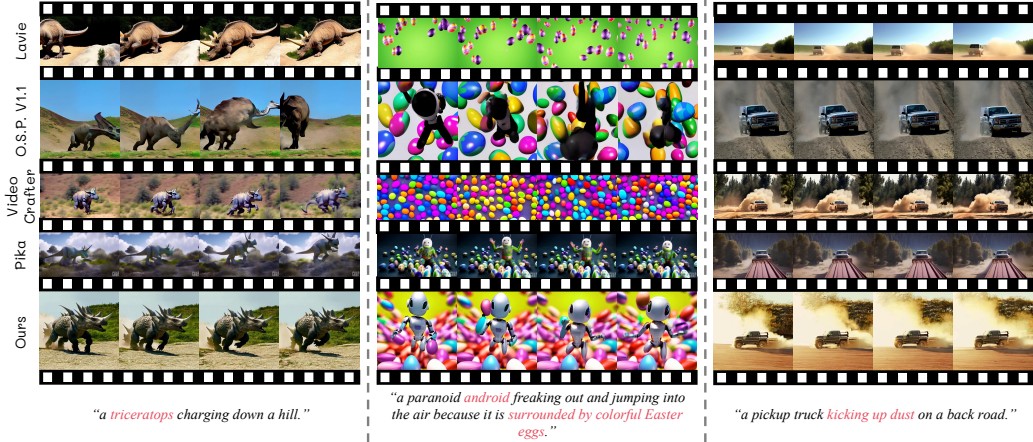

*"a triceratops charging down a hill."*      *"a paranoid android freaking out and jumping into the air because it is surrounded by colorful Easter eggs."*      *"a pickup truck kicking up dust on a back road."*

Figure 6: **Visual comparison of different T2V generation models.** Please zoom in for more details.

Table 5: Quantitative comparison of models trained on different datasets for **video restoration**.

| Method | Training Dataset | Dataset Size | PSNR↑ | SSIM↑ | LPIPS↓ | DOVER↑ | $E^*_{warp}$↓ |
|---|---|---|---|---|---|---|---|
| Upscale-A-Video (CVPR 2024) | WebVid, YouTube | ∼370K | 23.43 | 0.6195 | 0.2731 | 0.4863 | 0.00532 |
| Ours | OpenVidHD-0.4M | ∼130K | **23.49** | **0.7165** | **0.2015** | **0.5351** | **0.00283** |

| LR | Upscale-A-Video | Ours |
|---|---|---|

Figure 7: Visual comparison of restoring low-resolution (LR) video from UDM10 (Tao et al., 2017).

compared to closed-source commercial product Pika, which are trained on much larger datasets and with more computational resources. We present higher resolution versions in Figure 13, Figure 14 and Figure 15 for clearer comparison. Figure 5 presents a comprehensive analysis of the proposed model's performance against SOTA T2V models across various metrics.

**Video Restoration.** We further demonstrate the superiority of `OpenVidHD-0.4M` on the video restoration task. As shown in Table 5, we used `OpenVidHD-0.4M` to synthesize 130K training samples to train a video restoration model I2VGen-XL for arbitrary-resolution super-resolution, comparing it with the SOTA video restoration method, Upscale-A-Video (Zhou et al., 2024). The results show that our model outperforms across all metrics (both in fidelity and perception), demonstrating that `OpenVidHD-0.4M` significantly improves performance, even without task-specific design optimizations, due to its high quality. The visual comparison in Figure 7 shows that the model trained with `OpenVidHD-0.4M` produces clearer textures and more accurate structures.

### 6.3 ABLATION STUDY

**Ablations on Resolution, Architectures and Training Data.** Results are depicted in Table 6. We can draw the following conclusions: 1) Higher resolution leads to better metric scores. 2) The proposed MVDiT further improves both $VQA_A$ and $VQA_T$ compared to STDiT, indicating higher video quality and greater diversity. 3) More high-quality training data results in better metric scores.

**Ablations on MVDiT.** As shown in Table 7, we conduct ablations on MHCA module and scaling parameter $\alpha$ on MVDiT-256. From the results, we can draw the following conclusions: MHCA boosts video quality and alignment, and parameter $\alpha$ improves video quality and convergence. Notably, we observed that removing $\alpha$ causes the loss to decrease very slowly, indicating that $\alpha$ accelerates training, consistent with the findings reported in Peebles & Xie (2023). Please note that after removing MMTA, the model is unable to generate videos and instead produces multiple unrelated images, completely failing to meet the requirements for video generation.

Table 6: Ablations on different resolutions, architectures and training data. For models trained with 256×256 resolution, training iterations are selected at the similar steps for fair comparison. 'Pretrained Weight' means initializing with a corresponding pretrained model, *e.g.*, 'MVDiT-256' indicates that the MVDiT model with 256×256 resolution is used as the pretrained weight.

| Model | Resolution | Training Data | Pretrained Weight | VQA$_A$↑ | VQA$_T$↑ | Blip_bleu↑ | SD_score↑ | Clip_temp_score↑ | Warping_error↓ |
|---|---|---|---|---|---|---|---|---|---|
| STDiT | 256×256 | Ours-0.4M | PixArt-$\alpha$ | 11.11 | 12.46 | **24.55** | 67.96 | 99.81 | 0.0105 |
| STDiT | 512×512 | Ours-0.4M | STDiT-256 | 65.15 | 59.57 | 23.73 | 68.24 | 99.80 | 0.0089 |
| MVDiT | 256×256 | Ours-0.4M | PixArt-$\alpha$ | 22.39 | 14.15 | 23.72 | 67.73 | 99.71 | 0.0091 |
| MVDiT | 256×256 | OpenVid-1M | PixArt-$\alpha$ | 24.87 | 14.57 | 24.01 | 67.64 | 99.75 | 0.0081 |
| MVDiT | 512×512 | OpenVid-1M | MVDiT-256 | **66.65** | **63.96** | 24.14 | **68.31** | **99.83** | **0.0008** |

Table 7: Ablation studies on the effectiveness of modules in MVDiT.

| Setting | VQA$_A$↑ | VQA$_T$↑ | Blip_bleu↑ | SD_score↑ | Clip_temp_score↑ | Warping_error↓ |
|---|---|---|---|---|---|---|
| w/o MHCA | 13.9 | 12.35 | 19.74 | 67.58 | **99.73** | 0.0113 |
| w/o $\alpha$ | 3.16 | 3.55 | 14.38 | 66.94 | 99.01 | 0.0561 |
| MVDiT | **22.39** | **14.15** | **23.72** | **67.73** | 99.71 | **0.0091** |

Table 8: **Evaluation of human preference on captions** over $1,117$ samples and 10 volunteers.

| | Omission | Hallucinations | Distortion | Temporal Mismatch | Mean | Preference |
|---|---|---|---|---|---|---|
| Panda's short captions | 3.18 | **4.29** | 3.84 | 3.53 | 3.71 | 19.25% |
| Our generated long captions | **4.53** | 4.08 | **4.28** | **3.99** | **4.22** | **80.75%** |

**Human Preference on Captions.** In Table 8, we compare Panda's short captions with our generated long captions on $1,117$ validation samples, evaluated by 10 volunteers. Volunteers assessed captions across four criteria: (1) *Omission*: missing key elements, (2) *Hallucinations*: imagined elements, (3) *Distortion*: accuracy of attributes like color and size, and (4) *Temporal mismatch*: accuracy of event sequences. Each pair was rated from 1 to 5 (higher is better), and preferences were recorded. The results show: (1) The long captions provide richer descriptions, particularly in element accuracy and temporal events. (2) Though some hallucinations occur, accurate descriptions dominate. (3) Both captions can still be improved in modeling motion. (4) Long captions are strongly preferred overall.

# 7 CONCLUSION

In this work, we propose `OpenVid-1M`, a novel precise high-quality datasets for text-to-video generation. Comprising over 1 million high-resolution video clips paired with expressive language descriptions, this dataset aims to facilitate the creation of visually compelling videos. To ensure the inclusion of high-quality clips, we designed an automated pipeline that prioritizes aesthetics, temporal consistency, and fluid motion. Through clarity assessment and clip extraction, each video clip contains a single clear scene. Additionally, we curate `OpenVidHD-0.4M`, a subset of `OpenVid-1M` for advancing high-definition video generation. Furthermore, we propose a novel Multi-modal Video Diffusion Transformer (MVDiT) capable of achieving superior visually compelling videos, making full use of our powerful dataset. Extensive experiments and ablative analyses affirm the efficacy of `OpenVid-1M` compared to prior famous datasets, including WebVid-10M and Panda-50M.

**Limitations.** Despite advancements in T2V generation, our model, like previous SOTA models, also faces limitations in modeling the physical world. It sometimes struggles with intricate dynamics and motions of natural scenes, leading to unrealistic videos. We believe that with more high-quality training data, our model could be further scaled up and enhanced to handle such limitation.

# 8 ACKNOWLEDGEMENTS

This work was supported by Natural Science Foundation of China: No. 62406135, Natural Science Foundation of Jiangsu Province: BK20241198, the Gusu Innovation and Entrepreneur Leading Talents: No. ZXL2024362, and the AI & AI for Science Project of Nanjing University: No. 14380007.

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

## A   MORE IMPLEMENTATION DETAILS

### A.1   DATA PROCESSING PIPELINE

**Aesthetics and Clarity Assessment.**   We adopted the LAION Aesthetic Predictor and DOVER (Wu et al., 2023) to separately assess aesthetic and clarity scores due to their fast inference speeds and alignment with human preferences. These qualities make them efficient and well-suited for integration into our pipeline for processing million-level video data.

**Temporal Consistency.**   Extracting CLIP representations has proven effective for computing cosine similarity between images. We calculate the CLIP similarity (Radford et al., 2021) between every two adjacent frames in the video and take the average as an indicator of the temporal consistency, measuring the coherence and consistency of the video frames.

**Motion Difference.**   To measure motion amplitude, we utilize UniMatch (Xu et al., 2023), a pretrained state-of-the-art optical flow estimation method that is both efficient and accurate. We calculate the flow score between adjacent frames of the video, taking the squared average of the predicted values to represent motion dynamics, where higher values indicate stronger motion effects.

**Clip Extraction.**   Our observations reveal that fade-ins and fade-outs between consecutive scenes often go undetected when using a single cut detector with a fixed threshold. To address this, we employ a cascade of three cut detectors (Blattmann et al., 2023), each operating at different thresholds. This approach effectively captures sudden changes, fade-ins, and fade-outs in videos.

**Filtering Ratios.**   We randomly sampled a subset from the collected raw data and processed it through our data processing pipeline. A panel of evaluators was then tasked with assessing these video subsets, determining whether the videos at each processing step met our requirements. Based on their preferences, we derived score thresholds and filtering ratios for each step after multiple evaluations. Figure 8 provides visualizations of the videos with varying clarity, aesthetic, motion, and temporal consistency scores computed by our pipeline.

### A.2   DIFFERENCES BETWEEN MVDIT AND MMDIT

**Multi-Modal Self-Attention Module.**   We design a Multi-Modal Self-Attention (MMSA) module based on the self-attention module of MMDiT. To handle video data, we repeat the text tokens $T$ times and then concatenate the text tokens with video frame tokens using the same method as MMDiT. The self-attention operation is conducted along spatial and within the same frame. This provides a simple yet effective adaptation of MMDiT to video data input.

**Multi-Modal Temporal-Attention Module.**   Since MMDiT lacks the ability to generate videos, we introduce a *novel* Multi-Modal Temporal-Attention (MMTA) module on top of the MMSA module to efficiently capture temporal information in video data. To retain the advantages of the dual-branch structure in MMSA, we employ a similar approach in MMTA, incorporating a temporal attention layer to facilitate communication along the temporal dimension.

**Multi-Head Cross-Attention Module.**   Since the absence of semantic information may impair video generation performance, explicitly embedding semantic information from text tokens into visual tokens is helpful. To address this, we introduce a *novel* cross attention layer to enable direct communication between text and visual tokens.

## B   ABLATIONS ON DATA PROCESSING STEPS

We discuss effectiveness of each data processing step in `OpenVid-1M`, shown in Table 9: 1) Temporal screening improves Clip_temp_score and warping_error, enhancing temporal consistency. 2) Screening for aesthetics, temporal and motion boosts $VQA_A$, $VQA_T$, and Blip_bleu, suggesting better aesthetics and text understanding in generated videos. 3) Screening for clarity significantly improves $VQA_A$ and $VQA_T$. 4) Combining all four steps yields the highest scores in most metrics.

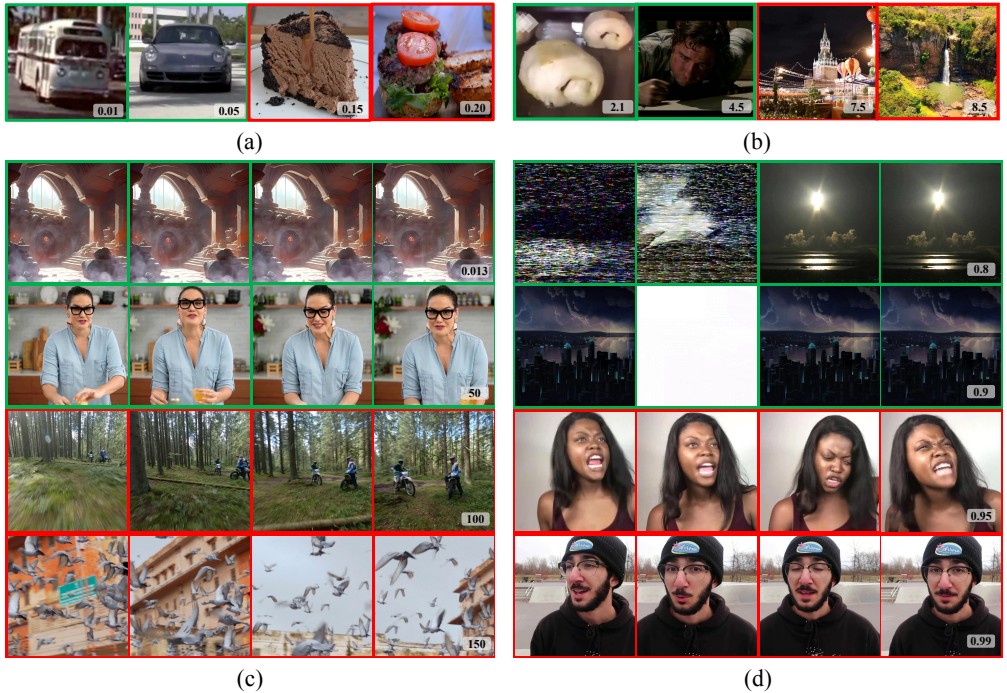

Figure 8: Visualizations of the videos with varying (a) clarity, (b) aesthetic, (c) motion, and (d) temporal consistency scores.

Table 9: Ablation studies on the effectiveness of each data processing step. The number of training data (0.6M), training iterations (50K) and resolution (256×256) for each setting are kept the same.

| Settings | | | | VQA$_A$↑ | VQA$_T$↑ | Blip_bleu↑ | SD_score↑ | Clip_temp_score↑ | Warping_error↓ |
| Aesthetics | Temporal | Motion | Clarity | | | | | | |
|---|---|---|---|---|---|---|---|---|---|
| ✔ | | | | 19.48 | 10.39 | 24.07 | 67.61 | 99.70 | 0.0137 |
| | ✔ | | | 20.40 | 10.90 | 23.31 | 67.57 | 99.73 | 0.0113 |
| | | ✔ | | 16.78 | 9.39 | 23.91 | 67.44 | 99.58 | 0.0217 |
| ✔ | ✔ | ✔ | | 20.32 | 11.42 | **24.43** | 67.62 | 99.71 | 0.0123 |
| ✔ | ✔ | ✔ | ✔ | **30.26** | **14.05** | 23.43 | **67.66** | **99.81** | **0.0081** |

## C  ACCELERATION FOR HD VIDEO GENERATION

Diffusion models, though powerful, often suffer from high computational costs and slow inference, especially for high-definition video generation. This is due to the sequential denoising process and attention computation, which has an $\mathcal{O}(L^2)$ complexity based on token length $L$. Inspired by Ma et al. (2024b), we observed significant temporal consistency in attention values between consecutive steps of the reverse denoising steps (Figure 9), revealing redundancy. These values can be cached and reused to accelerate denoising without retraining. Specifically, at timestep $t$, attention values are computed normally. At $t-1$, cached values for Self-Attention, Temporal-Attention, Cross-Attention, and Feedforward layers are reused, repeating this process every two steps. As shown in Figure 9, this method achieves up to a $1.7\times$ speedup at $1024$ resolution, with minimal quality impact. This indicates that diffusion models trained on `OpenVidHD-0.4M` can be accelerated efficiently without compromising quality.

## D  EXAMPLES OF `OpenVid-1M` DATASET

In Figure 10, we visualize some samples from our `OpenVid-1M`. We randomly select samples with $512 \times 512$ and $1920 \times 1080$ resolution, respectively. With well designed data processing pipeline, `OpenVid-1M` demonstrates superior quality and descriptive richness, particularly in aesthetics, motion, temporal consistency, caption length and clarity as well.

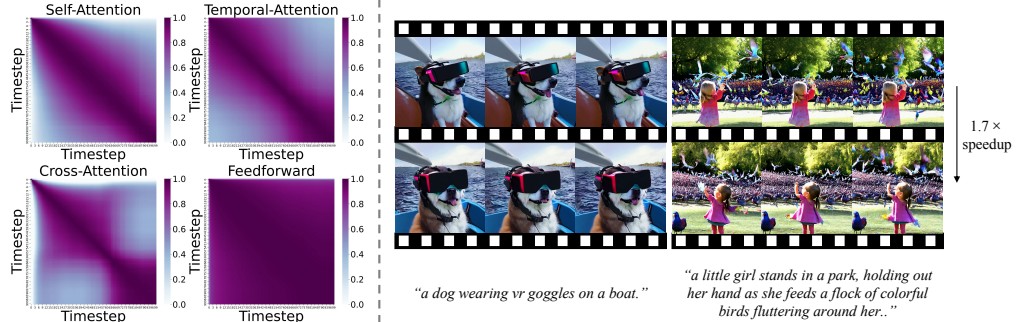

Figure 9: **Left:** Similarity for different attention values at different timesteps. **Right:** We compare the generation quality between accelerated model and original model at $1024$ resolution.

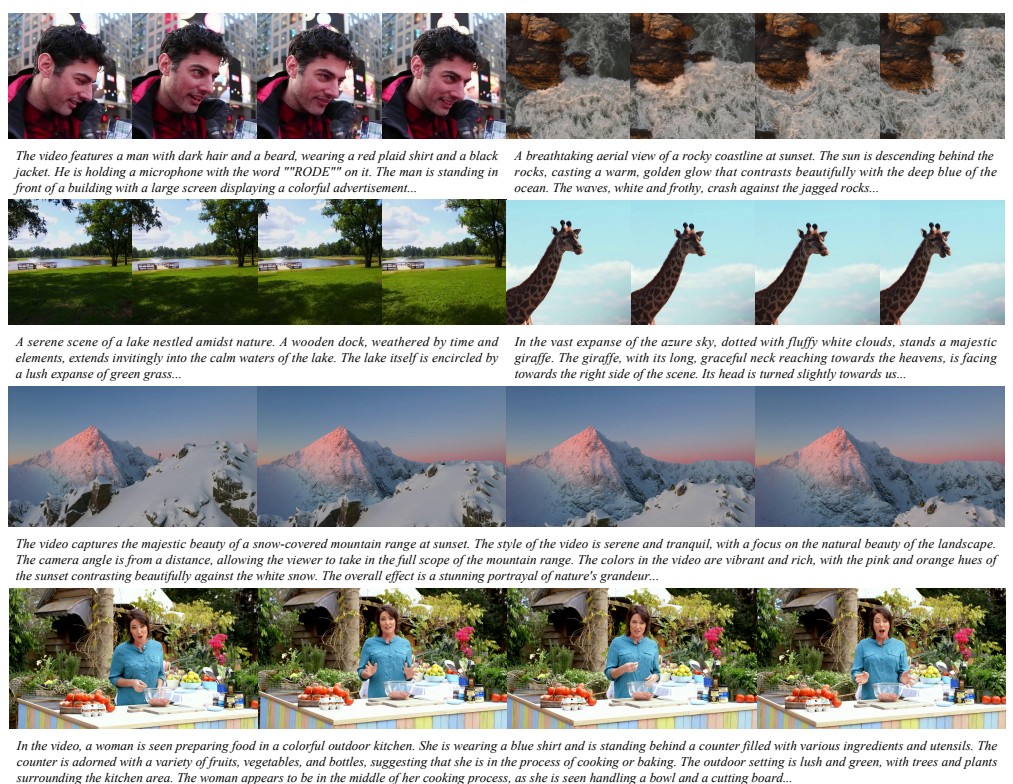

Figure 10: Examples of `OpenVid-1M` dsataset.

# E   MORE TEXT-TO-VIDEO EXAMPLES

We present more visual results of our model. As depicted in Figure 11, the first column illustrates our model's proficiency in generating aesthetically pleasing content with a painting style. The second column showcases the superior text alignment of the videos generated by our model, accurately depicting 'crashed down' from the text. The third and fourth columns highlight our model's ability to produce intricate dynamics and motions, e.g., 'motorcycle race' and 'gallop across'.

# F   VIDEO DURATIONS COMPARISON WITH OTHER DATASETS

We present video durations comparison between our `OpenVid-1M` and other million level text-to-video datasets in Figure 12. Specifically, `OpenVid-1M` consists of 1,019,957 clips, averaging 7.2 seconds each, with a total video length of 2,051 hours. Compared to previous million-level datasets, WebVid-10M contains low-quality videos with watermarks and Panda-70M contains many still, flickering, or blurry videos along with short captions. In contrast, our `OpenVid-1M` contains high-quality, clean videos with dense and expressive captions.

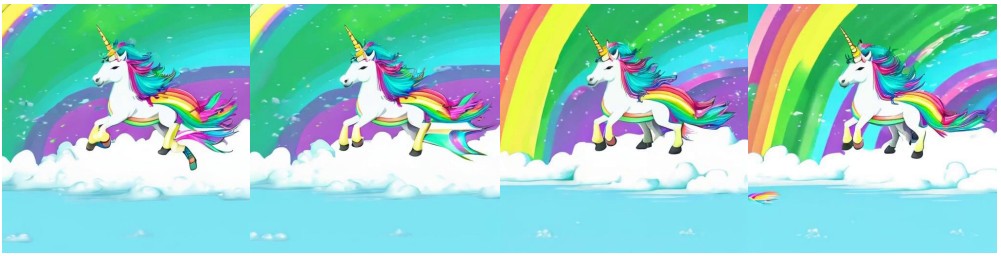

*"Unicorn sliding on a rainbow."*

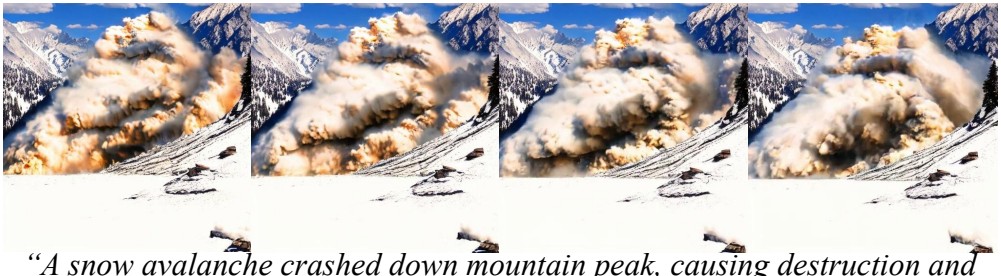

*"A snow avalanche crashed down mountain peak, causing destruction and mayhem."*

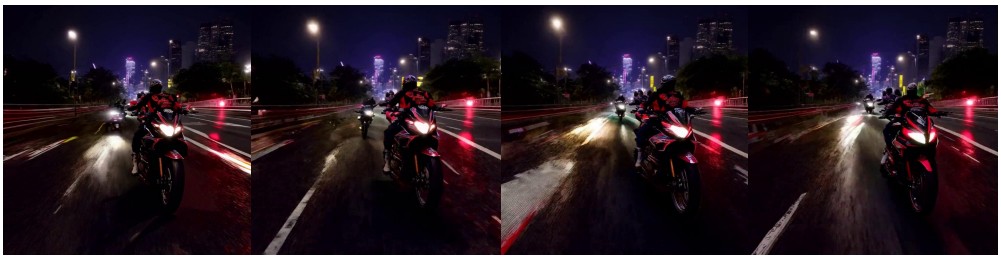

*"A motorcycle race through the city streets night."*

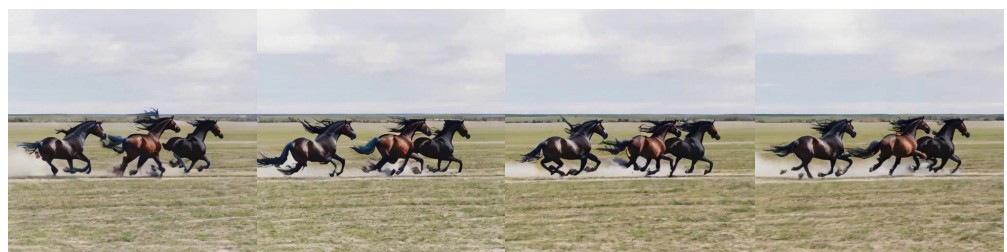

*"Three horses gallop across a wide open field, tails and manes flying in the wind."*

Figure 11: More text-to-video showcases.

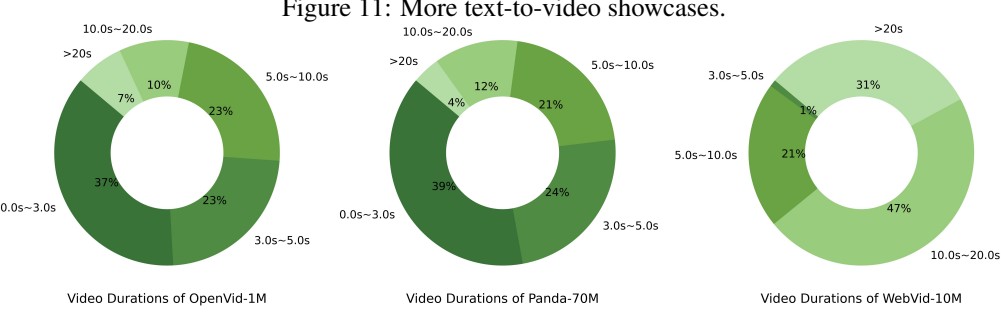

Figure 12: Comparisons on video durations with previous million level text-to-video datasets.

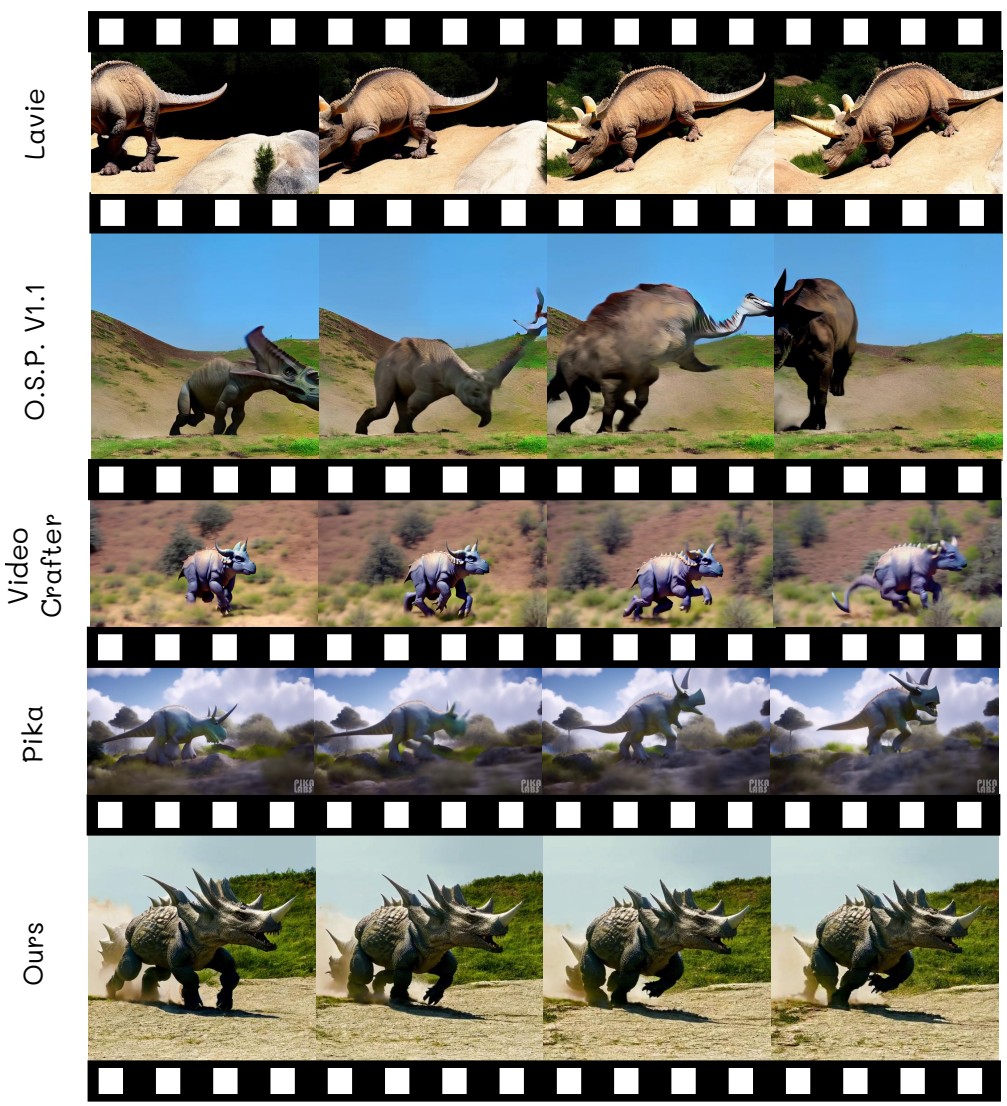

*"a triceratops charging down a hill."*

Figure 13: Visual comparison of different T2V generation models. Our model generates clearer, more aesthetically pleasing and more detailed videos due to our high-resolution `OpenVid-1M`.

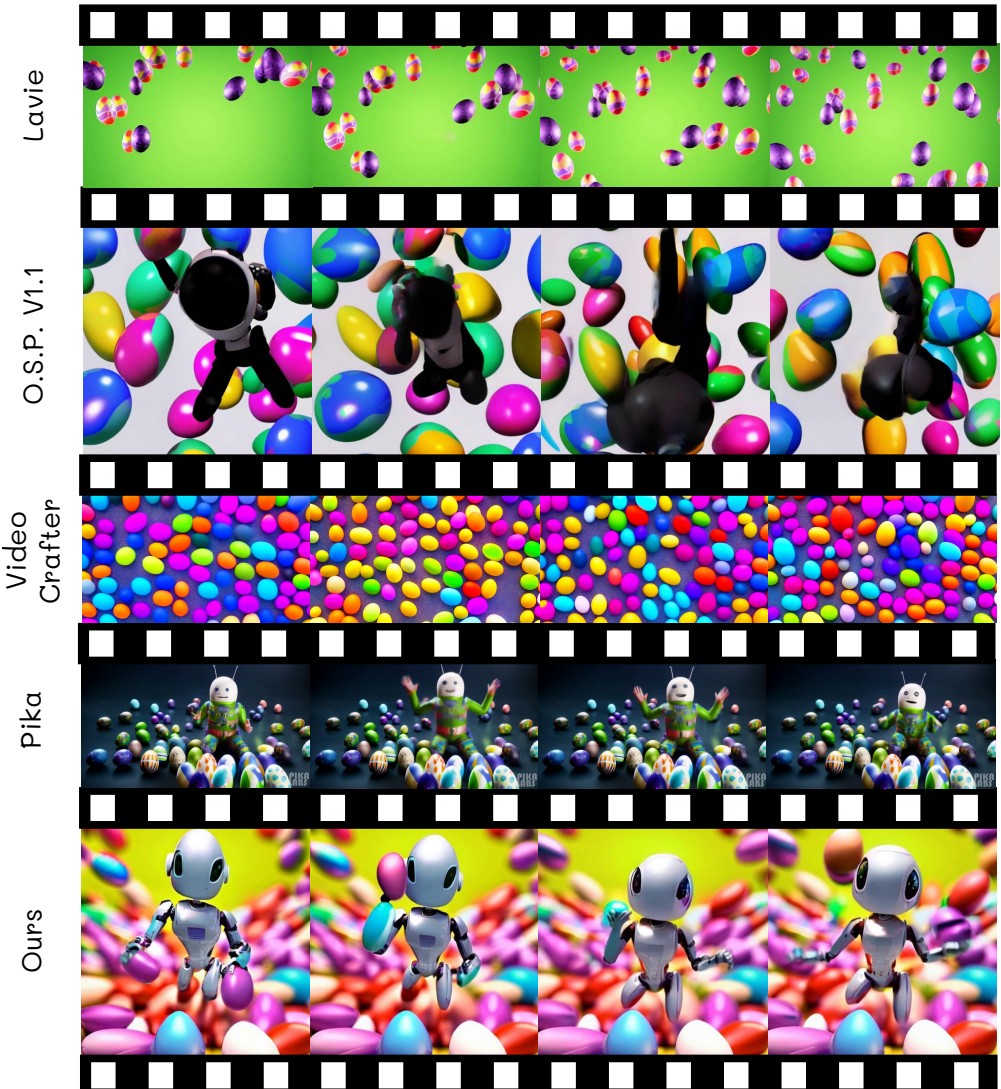

*"a paranoid android freaking out and jumping into the air because it is
surrounded by colorful Easter eggs."*

Figure 14: Visual comparison of different T2V generation models. Our model demonstrates a strong ability on prompt understanding, accurately depicting the 'android' and 'surrounded by colorful Easter eggs' from the text.

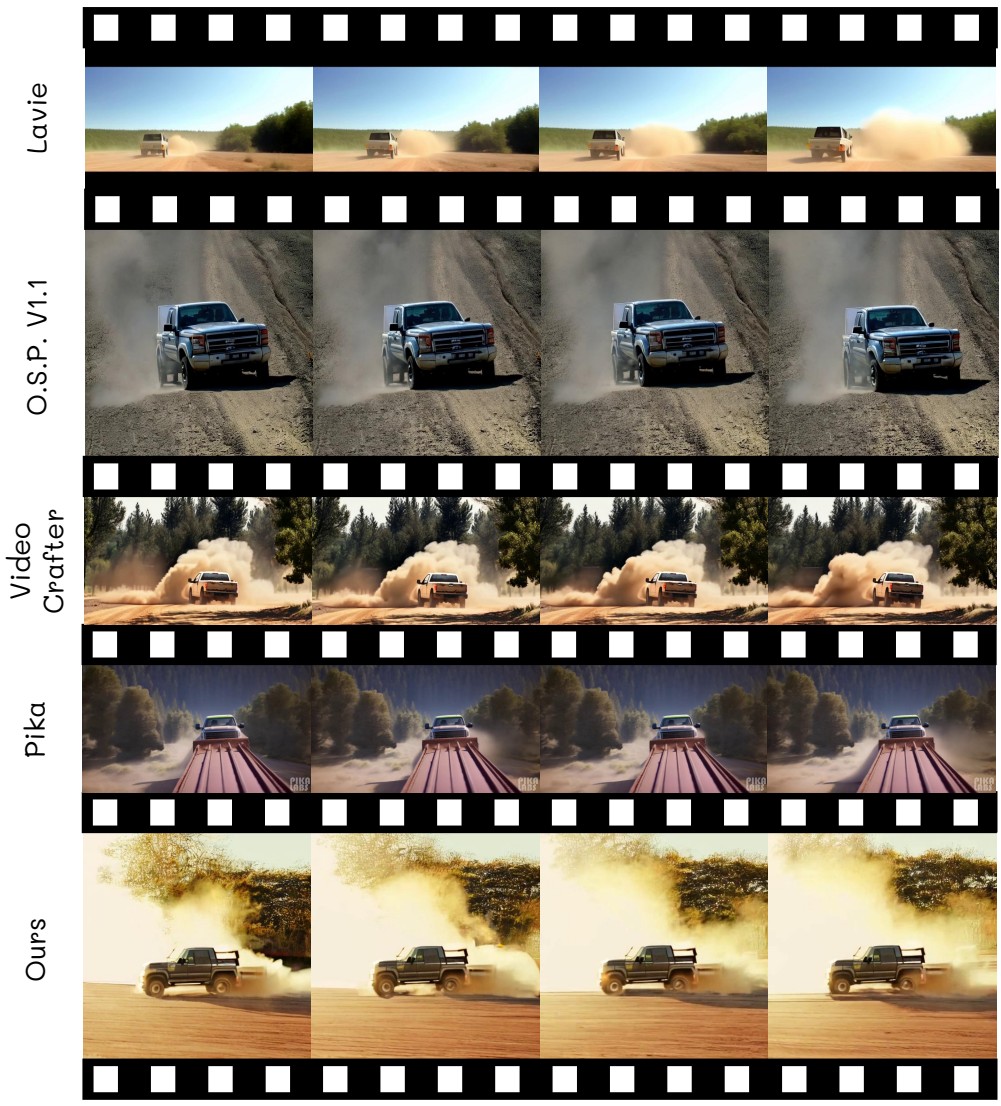

*"a pickup truck kicking up dust on a back road."*

Figure 15: Visual comparison of different T2V generation models. Our model better captures the 'kicking up dust', highlighting its superior motion quality.

