# OpenReview forum: "OpenVid-1M: A Large-Scale High-Quality Dataset for Text-to-video Generation"
_ICLR.cc/2025/Conference — ICLR 2025 Poster_

### Official Review · Reviewer_Kgx2 · 2024-10-17

**Soundness:** 3
**Presentation:** 3
**Contribution:** 4
**Rating:** 8
**Confidence:** 4

**Summary:**

This paper addresses two key issues in the text-to-video field: the lack of high-quality datasets and poor textual representations, proposing the 1M high-quality dataset OpenVid-1M. MVDiT is introduced to validate the effectiveness of the dataset.

**Strengths:**

1. The paper resolves the critical issue of the lack of high-quality datasets in the text-to-video field, significantly impacting the fine-tuning and training of pre-trained text-to-video models, especially smaller models.

2. The article presents two datasets of different scales: 1M and 0.4M, with the 1M dataset having a resolution greater than 512.

3. The dataset’s text is highly detailed, making it suitable for future transformer or DiT-based video training models that require long text inputs.

4. The article selects high-quality data from multiple datasets and various models, demonstrating significant effort.

5. Experiments training the same model on different datasets indicate that this dataset is indeed of high quality and can enhance the model's output performance.

6. The article has a clear motivation, with writing that is clear and easy to understand.

**Weaknesses:**

I tend to rate the paper between 6 to 8 points, but I'm currently giving it a score of 6. Despite its rich experiments and significant contributions, there are still the following issues:

1. For the ICLR conference, this paper lacks explanatory work, such as a clear justification for each step of the dataset filtering process. Specifically, it does not adequately explain why certain models were chosen or the rationale behind the selected filtering ratios.

2. OpenVid-1M only filters and integrates existing datasets and does not include any new high-quality videos. Models trained on this dataset do not learn new knowledge.

3. Using LLaVA to generate captions for videos merely indicates that the captions are longer and does not guarantee improved accuracy or richness of the descriptions compared to the originals. Models trained with such captions primarily transfer some knowledge from LLaVA rather than gaining new knowledge to achieve performance breakthroughs.

**Questions:**

1. Why choose the top 20% of Panda-50M while selecting the top 90% from other datasets?

2. How is optical flow used to filter videos? Why is temporal consistency used to discard the highest and lowest, while optical flow is not?

3. In Table 3, is the comparison across different resolutions fair? Could super-resolution impact the quality of the original videos?

4. Is the dataset publicly available?

**Details Of Ethics Concerns:**

The article will make the dataset publicly available, and the text information in the dataset is AI-generated, requiring a review of the textual data. The video information in the dataset is selected from publicly available datasets using specific strategies, so it should not require further review.

---

> ### Author Response · Authors · 2024-11-23
> **Response to Reviewer Kgx2 (1/2)**
>
> Thank you for recognizing our work and providing valuable and detailed suggestions. We sincerely appreciate your recognition of our dataset's impact, the significance of our efforts, and the clarity of our motivation and writing! We have carefully considered your constructive and insightful comments, and here are our responses to your concerns.
>
> **Q1: Weaknesses - Lacking explanatory work.**
>
> **(1) Filtering models:**
>
> *Aesthetics and Clarity Assessment:* We adopted the LAION Aesthetic Predictor and DOVER to separately assess aesthetic and clarity scores due to their fast inference speeds and alignment with human preferences. These qualities make them efficient and well-suited for integration into our pipeline for processing million-level video data.
>
> *Temporal Consistency:* Extracting CLIP representations has proven effective for computing cosine similarity between images. We calculate the CLIP similarity between every two adjacent frames in the video and take the average as an indicator of the temporal consistency, measuring the coherence and consistency of the video frames.
>
> *Motion Difference:* To measure motion amplitude, we utilize UniMatch, a pretrained state-of-the-art optical flow estimation method that is both efficient and accurate. We calculate the flow score between adjacent frames of the video, taking the squared average of the predicted values to represent the motion dynamics, where higher values indicate stronger motion effects.
>
> *Clip Extraction:* Our observations reveal that fade-ins and fade-outs between consecutive scenes often go undetected when using a single cut detector with a fixed threshold. To address this, we employ a cascade of three cut detectors, each operating at different thresholds. This approach effectively captures sudden changes, fade-ins, and fade-outs in videos.
>
> **(2) Filtering ratios**
>
> We randomly sampled a subset from the collected raw data and processed it through our data processing pipeline. A panel of evaluators was then tasked with assessing these video subsets, determining whether the videos at each processing step met our requirements. Based on their preferences, we derived score thresholds and filtering ratios for each step after multiple evaluations. Figure 2 in supplementary material provides visualizations of the videos with varying clarity, aesthetic, motion, and temporal consistency scores computed by our pipeline.
>
> **Q2: Weaknesses -  Models trained on dataset do not learn new knowledge.**
>
> Previous datasets contain a significant amount of *noisy* data, which severely impacts the performance of models trained on them. As demonstrated in Table 7(*Paper*) and the additional Panda-50M-HD experiments in **Q6-2)**, this issue is evident. Therefore, constructing the high-quality OpenVid-1M dataset is both meaningful and essential to prevent models from learning *useless or harmful* knowledge. Furthermore, our dataset is also valuable in other research tasks, such as video super-resolution or frame interpolation, which currently lack large-scale, high-quality video datasets like OpenVid-1M.

---

> ### Author Response · Authors · 2024-11-23
> **Response to Reviewer Kgx2 (2/2)**
>
> **Q3: Weaknesses - Captions generated by LLaVA.**
>
> 1)In Table 8(*Paper*), we compared Panda’s short captions with our generated long captions on 1, 117 validation samples, evaluated by 10 volunteers. The results demonstrate that our generated captions provide *richer* descriptions, particularly in element accuracy and temporal events. While some hallucinations are present, accurate descriptions are predominant.
>
> 2)We realized the issue of LLaVA on hallucinations and made several attempts to further improve the captions.
>
> Specifically, we incorporated two additional long video captioners: 1) ShareGPT4Video. A recent model trained on captions generated by GPT-4V; and 2) LLaVA-Video-72B-Improved (Ours). Given the high cost of captioning large volumes of videos with commercial products, we developed an improved chain-of-thought pipeline based on the recent powerful LLaVA-Video-72B model. This approach enhances instance-level detail in the videos and reduces hallucinations by segmenting and analyzing objects individually. Notably, LLaVA-Video-72B achieves comparable performance to GPT-4V and Gemini on the video understanding benchmark [1].
>
> We carefully selected 22K videos from OpenVid-1M to recaption using LLaVA-V1.6-34B (Ours-in-submission), ShareGPT4Video, and LLaVA-Video-72B-Improved (Ours-new). We fine-tuned STDiT-256 model using these three subsets, and the experimental results are shown in the table below.
>
> |  | VQA$_A$$\uparrow$ | VQA$_T$$\uparrow$ | Blip Bleu$\uparrow$ | SD Score$\uparrow$ | Clip Temp Score$\uparrow$ | Warping Error$\downarrow$ |
> |:------------:|:------:|:------:|:------:|:------:|:------:|:------:|
> | LLaVA-V1.6-34B | 16.08 | 14.61 | 24.05 | 68.34 | 99.73 | **0.0093** |
> | ShareGPT4Video | 16.29 | **15.74** | 24.07 | 68.31 | 99.71 | 0.0099 |
> | LLaVA-Video-72B-Improved | **16.89** | 14.86 | **24.24** | **68.39** | **99.79** | 0.0096 |
>
> The table shows that training on the dataset with our fine-grained captions leads to improved performance on alignment.
> The results will be included in the final version. Moving forward, we will continue to explore video captioning methods and collect higher-resolution, longer-duration video data to further enhance OpenVid-1M.
>
> [1] Fu C, Dai Y, Luo Y, et al. Video-mme: The first-ever comprehensive evaluation benchmark of multi-modal llms in video analysis[J]. arXiv preprint arXiv:2405.21075, 2024.
>
> **Q4: Questions - Filtering ratios.**
>
> Please refer to **Q1** for more details.
>
> **Q5: Questions - Optical flow used in Motion Difference.**
>
> 1) Please refer to **Q1** for filtering process details.
>
> 2) We retain clips with moderate flow scores and filter out the highest and lowest like temporal consistency. We will make it clearer in the revised version.
>
> **Q6. Questions - Fair comparison across different models**
>
> 1)Since previous methods support varying resolutions, performance comparisons cannot be entirely consistent. However, in our paper, we presented results for our model at 512 and 1024 resolutions, which use significantly less training data compared to most previous methods (with similar resolutions) while achieving superior performance.
>
> 2)To address your concerns regarding super-resolution, we selected 1080P videos from Panda-50M to create Panda-50M-HD for a fair comparison (as WebVid-10M does not include high-resolution videos). The table below compares the performance of models trained on OpenVid-1M and Panda-50M-HD.
>
> | Resolution | Training Data | VQA$_A$$\uparrow$ | VQA$_T$$\uparrow$ | Blip Bleu$\uparrow$ | SD Score$\uparrow$ | Clip Temp Score$\uparrow$ | Warping Error$\downarrow$ |
> |:------------:|:------:|:------:|:------:|:------:|:------:|:------:|:------:|
> | 1024×1024 | Panda-50M (4× Super-resolution) | 63.25 | 53.21 | **23.60** | 67.44 | 99.57 | 0.0163 |
> | 1024×1024 | Panda-50M-HD | 13.48 | 42.89 | 21.78 | **68.43** | 99.84 | 0.0136 |
> | 1024×1024 | OpenVidHD-0.4M (Ours) | **73.46** |  **68.58** | 23.45 | 68.04 | **99.87** | **0.0052** |
>
> From the table, we observe that the model trained on our OpenVid-1M demonstrates superior performance, while the model trained on Panda-50M-HD performs poorly. This discrepancy may be attributed to the low-quality videos in Panda-50M-HD (e.g., low aesthetics and clarity, nearly static scenes, and frequent flickering), a problem our data processing pipeline effectively avoids.
>
> **Q7. Questions - Is the dataset publicly available?**
>
> We will fully open-source our codes, models, the OpenVid-1M dataset and the complete data processing workflow to facilitate the community.

---

> > ### Comment · Reviewer_Kgx2 · 2024-11-26
> >
> > Thanks, your answer solved my problem

---

> > > ### Author Response · Authors · 2024-11-26
> > > **Response to Reviewer Kgx2**
> > >
> > > Thank you very much for your feedback and for raising the score! We’re delighted to hear that our reply solved your problem. We also truly appreciate your thorough and valuable feedback, which has greatly helped improve our work.

---

### Official Review · Reviewer_Y42k · 2024-10-21

**Soundness:** 3
**Presentation:** 3
**Contribution:** 3
**Rating:** 6
**Confidence:** 4

**Summary:**

This paper proposes a large-scale high-quality dataset for text-to-video generation, named OpenVid-1M. Compared to existing related datasets, OpenVid-1M has the edge in high-quality videos and expressive captions. OpenVid-1M is curated from ChronoMagic, CelebvHQ, Open-Sora-plan and Panda by controlling aesthetic score, temporal consistency, motion difference, clarity assessment, clip extraction and video caption. Besides, the authors follow MMDiT and design a Multi-modal Video Diffusion Transformer (MVDiT) architecture for text-to-video generation.

**Strengths:**

1. The authors descrive the advantages of the proposed dataset clearly.
2. There are sufficient experiments to evaluate the dataset and model.

**Weaknesses:**

1. Since the proposed MVDiT follows MMDiT, the authors should compare the differences between the two in detail through text descriptions or figures. For example, what modules does MVDiT retain, remove, or add from MMDiT? How do these changes more effectively cope with video data?
2. As a work on text-to-video generation, there is no project or demo webpage to showcase the dataset or model performance. It is difficult to judge the overall quality of the video from a few frames captured from the video, so I would like to know if the author has plans to make the demo website public.
3. It would be better to compare the video duration distribution between OpenVid-1M and other datasets in the form of Figure 3 Right, since the video duration can influence the quality of video generation.

**Questions:**

See weaknesses.

---

> ### Author Response · Authors · 2024-11-23
> **Response to Reviewer Y42k**
>
> Thank you for recognizing our work and providing valuable suggestions. **We will fully open-source our codes, models, the OpenVid-1M dataset and the complete data processing workflow to support the community.** We have carefully considered your constructive and insightful comments and here are our responses to your concerns.
>
> **Q1. Differences between MVDiT and MMDiT.**
>
> Thank you for your practical suggestion. Below, we outline the differences between our MVDiT and MMDiT:
>
> *1) Multi-Modal Self-Attention:* We design a Multi-Modal Self-Attention (MMSA) module based on the self-attention module of MMDiT. To handle video data, we repeat the text tokens T times and then concatenate the text tokens with video frame tokens using the same method as MMDiT. This provides a simple yet effective adaptation of MMDiT to video data input.
>
> *2) Multi-Modal Temporal-Attention:* Since MMDiT lacks the ability to generate videos, we introduce a *novel* Multi-Modal Temporal-Attention (MMTA) module on top of the MMSA module to efficiently capture temporal information in video data. To retain the advantages of the dual-branch structure in MMSA, we employ a similar approach in MMTA, incorporating a temporal attention layer to facilitate communication along the temporal dimension.
>
> *3) Multi-Head Cross-Attention:* Since the absence of semantic information may impair video generation performance, explicitly embedding semantic information from text tokens into visual tokens is helpful. To address this, we introduce a *novel* Cross-Attention Layer to enable direct communication between text and visual tokens.
>
> Additionally, to validate the effectiveness of our proposed MMTA and MMCA, we conducted ablation studies on MVDiT-256. The results are presented in the table below.
>
> |  | VQA$_A$$\uparrow$ | VQA$_T$$\uparrow$ | Blip Bleu$\uparrow$ | SD Score$\uparrow$ | Clip Temp Score$\uparrow$ | Warping Error$\downarrow$ |
> |:------------:|:------:|:------:|:------:|:------:|:------:|:------:|
> | w/o MMTA | / | / | / | / | / | / |
> | w/o MMCA | 13.9 | 12.35 | 19.74 | 67.58 | **99.73** | 0.0113 |
> | MVDiT | **22.39** | **14.15** | **23.72** | **67.73** | 99.71 | **0.0091** |
>
> From the table, we can find that MMCA boosts video quality and alignment. Please note that after removing MMTA, the model is *unable* to generate videos and instead produces multiple unrelated images, completely failing to meet the requirements for video generation. We will include these results and analyses in the revised version.
>
> Last but not least, *our architecture design of MVDiT* received positive feedback from Reviewer zA3J.
>
> We will include these results and analyses in the revised version.
>
> **Q2. Demo website.**
>
> We are in the process of developing a demo webpage, which will be made publicly accessible upon completion. To address your concerns, we have included additional visualizations of our OpenVid-1M dataset and generated video results in Figure 1 and Figure 3 of the supplementary material. These results will also be incorporated into the appendix of the revised version.
>
> **Q3. Video duration distribution comparison.**
>
> Thank you for your valuable feedback. Per your request, we compared the video duration distribution using annular charts, with the results presented in Figure 4 of the supplementary material. We will include this comparison in the revised version.

---

> > ### Comment · Reviewer_Y42k · 2024-11-26
> > **Response to Authors**
> >
> > Thanks to the authors for the detailed rebuttal. It is nice to clarify the differences between MVDiT and MMDiT. Besides, additional visualizations of generated video results are good. It would be better to give some GIFs to present the quality and consistency of the video. All things considered, I want to maintain my score and look forward to the revised version.

---

> > > ### Author Response · Authors · 2024-11-27
> > > **Response to Reviewer Y42k**
> > >
> > > Thank you very much for your response! We’re delighted to hear that our reply solved your problem. We also truly appreciate your thorough and valuable feedback, which has greatly helped improve our work.

---

### Official Review · Reviewer_e1N1 · 2024-10-28

**Soundness:** 3
**Presentation:** 3
**Contribution:** 3
**Rating:** 8
**Confidence:** 4

**Summary:**

This paper introduce OpenVid-1M dataset, a new precise high-quality datasets for T2V generation. This dataset consists of about 1M videos, all with resolutions of at least 512x512, accompanied by detailed and long captions, facilitating the creation of visually compelling videos. An automated filtering and annotation pipeline is proposed to ensure high-quality of the dataset. Additionally, a new Multi-modal Video Diffusion Transformer (MVDiT) method is proposed to incorporate multi-modal information for better visual quality. Extensive experimental results verify the effectiveness of the proposed dataset and method.

**Strengths:**

1.	A dataset is proposed, comprising over 1 million high-resolution video clips paired with expressive language descriptions, this dataset aims to facilitate the creation of visually compelling videos.
2.	An automated filtering and annotation pipeline is proposed to ensure the quality of videos.
3.	Extensive experimental results verify the effectiveness of the proposed method.

**Weaknesses:**

1.	The proposed automatic data cleaning pipeline seems to be a pipeline that many previous methods have commonly used in SD-3 and SVD, lacking a certain novelty.
2.	Lack of ablation study for the proposed method, such as the effectiveness of scaling parameter α and Multi-Modal Temporal-Attention Module.
3.	The video shown in Figure 6 and Figure 8 takes up very little space, making it difficult to see the details clearly. Increasing the size of the video frames or providing higher resolution versions in an appendix would be helpful.
4.	The section on Acceleration for HD Video Generation seems redundant and is not the method proposed in this work. It can be placed in the appendix, leaving room for more qualitative text-to-video results.

**Questions:**

Refer to Weaknesses for more details.

---

> ### Author Response · Authors · 2024-11-23
> **Response to Reviewer e1N1**
>
> Thank you so much for acknowledging the strength of our method. We have carefully considered your constructive and insightful comments and here are our responses to your concerns.
>
> **Q1. Novelty of data processing pipeline.**
>
> The SD3 pipeline focuses on collecting *image* training data, whereas our pipeline incorporates steps such as Temporal Consistency, Motion Difference, and Clip Extraction, specifically designed to address challenges unique to *video* data. Therefore, our approach is fundamentally different from SD3.
>
> The differences between our pipeline and SVD's have already been discussed in Section 4 of the paper. However, we would like to reiterate and clarify them here:
>
> **1) Visual quality evaluation:** We integrate DOVER for assessing video clarity, which provides better texture and clarity analysis compared to SVD's CLIP aesthetic scoring.
>
> **2) Motion evaluation:** We adopt UniMatch to replace traditional optical flow methods, such as Farneback and RAFT, enabling more accurate processing of both static and fast-moving videos while maintaining computationally efficient.
>
> **3) Time consistency evaluation:** Beyond clip extraction to avoid abrupt changes, we additionally incorporate flicker removal to improve temporal consistency.
>
> **4) Processing efficiency:** Instead of extracting clips from a large video pool as done in SVD, we pre-filter high-quality videos, significantly reducing redundant processing.
>
> **5) Open-sourcing:** In contrast to SVD's *unreleased* data processing code and dataset, we will *release* our codes, pretrained models, the OpenVid-1M dataset and the complete data processing workflow to benefit the community.
>
> **Q2. Ablation study of MVDiT.**
>
> Per your request, the table below presents ablation studies validating the effectiveness of the scaling parameter α, Multi-Modal Temporal-Attention (MMTA) and Multi-Modal Cross-Attention (MMCA) layers on MVDiT-256.
>
> |  | VQA$_A$$\uparrow$ | VQA$_T$$\uparrow$ | Blip Bleu$\uparrow$ | SD Score$\uparrow$ | Clip Temp Score$\uparrow$ | Warping Error$\downarrow$ |
> |:------------:|:------:|:------:|:------:|:------:|:------:|:------:|
> | w/o MMTA | / | / | / | / | / | / |
> | w/o MMCA | 13.9 | 12.35 | 19.74 | 67.58 | **99.73** | 0.0113 |
> | w/o α | 3.16 | 3.55 | 14.38 | 66.94 | 99.01 | 0.0561 |
> | MVDiT | **22.39** | **14.15** | **23.72** | **67.73** | 99.71 | **0.0091** |
>
> From the table, we can draw the following conclusions: MMCA boosts video quality and alignment, and parameter α improves video quality and convergence. Notably, we observed that removing α causes the loss to decrease very slowly, indicating that α accelerates training, consistent with the findings reported in [1]. Please note that after removing MMTA, the model is *unable* to generate videos and instead produces multiple unrelated images, completely failing to meet the requirements for video generation. These results and analyses will be included in the final version.
>
> [1] Peebles W, Xie S. Scalable diffusion models with transformers[C]//Proceedings of the IEEE/CVF International Conference on Computer Vision. 2023: 4195-4205.
>
> **Q3. Clearer qualitative evaluation.**
>
> Thank you for your valuable suggestion. In the revised version, we will enlarge the video frames in Figures 6 and 8 and provide higher-resolution versions in the appendix to improve clarity.
>
> **Q4. The section on Acceleration.**
>
> Thank you for your valuable feedback. In the revised version, we will move this section to the appendix and replace it with more qualitative text-to-video results.

---

> > ### Comment · Reviewer_e1N1 · 2024-11-27
> >
> > Thank the authors for the responses. My concerns are solved. And I decide to maintain my score rate.

---

> > > ### Author Response · Authors · 2024-11-27
> > > **Response to Reviewer e1N1**
> > >
> > > Thank you very much for your response! We’re delighted to hear that our reply solved your concerns. We also truly appreciate your thorough and valuable feedback, which has greatly helped improve our work.

---

### Official Review · Reviewer_zA3J · 2024-11-04

**Soundness:** 3
**Presentation:** 2
**Contribution:** 3
**Rating:** 6
**Confidence:** 4

**Summary:**

This paper proposes a high-quality large-scale T2V dataset namely OpenVid-1M, as well as a new T2V model architecture – the MVDIT.

As main contribution, OpenVid-1M is born out of several open sourced dataset such as Panda, CelebvHQ, etc. and filtered with carefully designed pipeline followed by recaption.

As secondary contribution, the proposed MVDIT can be considered as a straight extension of MMDIT, where video and text token are jointly feed to 3 successive attention modules. The author(s) claim such design can mine structure information from visual feature and semantic information form text feature, and verify it through experiments.

**Strengths:**

1. Meticulously designed data process pipeline, which producing relatively higher quality comparing to previous datasets. There is no doubt that a publicly available million level dataset with high quality is critical for video generation task.


2. The architecture designing of MVDIT is reasonable. Text tokens are repeated by T times to fitting the frames, which makes it natural to equally treat visual and semantic information in the self-attention and temporal-attention modules.


3. Superior generation results among popular open sourced T2V systems. BTW, is it possible to make the trained model released?

**Weaknesses:**

1.	As a dataset paper. The proposed OpenVid-1M is somewhat weak. First, it is in fact a downstream collection of several publicly available datasets, which doesn’t provide extra videos. (Are you considering collecting new video data?)  Second，in contrast with carefully designed filtering operations, it is too crude to directly use raw LLAVA model as captioner without any comparison, since video caption is extremely important for T2V task. It is suggested to try sophisticated commercial LMMs such as GPT 4V and Gemini, or to finetune task aware open source models.

2.	The introduced MVDIT is greatly inspired by MMDIT. It can be seen as a naturally extension to T2V task of MMDIT, which largely limit its technic novelty. It is notable in Figure 4 that this work adds Temporal-Attention and Cross-Attention layers besides Self-Attention in MMDIT,  so can you take an empirical ablation study to verify their effectiveness?

**Questions:**

1.	In Figure 3 Right, it is noticed that 37% clips are less than 3s. Considering mainstream T2V  systems are more than 5s, is it valuable to keep such short clips? Can you perform an ablation study showing the impact of filtering out clips shorter than 5 seconds on model performance?
2.	Is the Self-Attention Module along spatial and within the same frame? I guess so. Please make it clearer. Furthermore, have you tried full 3D attention( Open Sora Plan v1.2) with self attention module?
3.	In table 3, there are some competitive models appearing before submission DDL of ICLR, such as CogvideoX-5B and OpenSoraPlanV1.2. Can you add them in the SOTA list?
4.	In table 4, to compare model trained by OpenVidHD with the one 4x by Panda 50M is not fair. Is it possible to select 1020P from Panda 50M to form Panda 50MHD for fair comparison?

---

> ### Author Response · Authors · 2024-11-23
> **Response to Reviewer zA3J (1/3)**
>
> Thank you for recognizing our work and providing valuable and detailed suggestions. We are pleased that you recognized our dataset to be *critical*, our architecture to be *reasonable*, and our generation results to be *superior*! **We will fully open-source our codes, models, the OpenVid-1M dataset and the complete data processing workflow to support the community.**
>
> **Weaknesses:**
>
> **Q1: The proposed OpenVid-1M is somewhat weak.**
>
> Thanks for your suggestions. We realized this issue and made several attempts, especially on improving the captions and enriching the comparison among different captioners in addition to the user study between long captions (Ours) and short ones (Panda) shown in Table 8(Paper).
>
> As you suggested, we incorporated two additional long video captioners: 1) *ShareGPT4Video*. A recent model trained on captions generated by GPT-4V; and 2) *LLaVA-Video-72B-Improved (Ours)*. Given the high cost of captioning large volumes of videos with commercial products, we developed an improved chain-of-thought pipeline based on the recent powerful LLaVA-Video-72B model. This approach enhances instance-level detail in the videos and reduces hallucinations by segmenting and analyzing objects individually. Notably, LLaVA-Video-72B achieves comparable performance to GPT-4V and Gemini on the video understanding benchmark [1].
>
> Specifically, we carefully selected 22K videos from OpenVid-1M to recaption using LLaVA-V1.6-34B (Ours-in-submission), ShareGPT4Video, and LLaVA-Video-72B-Improved (Ours-new). We fine-tuned STDiT-256 model using these three subsets, and the experimental results are shown in the table below.
>
> |  | VQA$_A$$\uparrow$ | VQA$_T$$\uparrow$ | Blip Bleu$\uparrow$ | SD Score$\uparrow$ | Clip Temp Score$\uparrow$ | Warping Error$\downarrow$ |
> |:------------:|:------:|:------:|:------:|:------:|:------:|:------:|
> | LLaVA-V1.6-34B | 16.08 | 14.61 | 24.05 | 68.34 | 99.73 | **0.0093** |
> | ShareGPT4Video | 16.29 | **15.74** | 24.07 | 68.31 | 99.71 | 0.0099 |
> | LLaVA-Video-72B-Improved | **16.89** | 14.86 | **24.24** | **68.39** | **99.79** | 0.0096 |
>
> The table shows that training on the dataset with our fine-grained captions leads to improved performance on alignment. The results will be included in the final version. Moving forward, we will continue to explore video captioning methods and collect higher-resolution, longer-duration video data to further enhance OpenVid-1M.
>
> [1] Fu C, Dai Y, Luo Y, et al. Video-mme: The first-ever comprehensive evaluation benchmark of multi-modal llms in video analysis[J]. arXiv preprint arXiv:2405.21075, 2024.
>
> **Q2: Ablation study on MVDIT.**
>
> The table below presents ablation studies validating the effectiveness of the Multi-Modal Temporal-Attention (MMTA) and Multi-Modal Cross-Attention (MMCA) layers on MVDiT-256.
>
> |  | VQA$_A$$\uparrow$ | VQA$_T$$\uparrow$ | Blip Bleu$\uparrow$ | SD Score$\uparrow$ | Clip Temp Score$\uparrow$ | Warping Error$\downarrow$ |
> |:------------:|:------:|:------:|:------:|:------:|:------:|:------:|
> | w/o MMTA | / | / | / | / | / | / |
> | w/o MMCA | 13.9 | 12.35 | 19.74 | 67.58 | **99.73** | 0.0113 |
> | MVDiT | **22.39** | **14.15** | **23.72** | **67.73** | 99.71 | **0.0091** |
>
> From the table, we can find that MMCA boosts video quality and alignment. Please note that after removing MMTA, the model is *unable* to generate videos and instead produces multiple unrelated images, completely failing to meet the requirements for video generation.
>
> These results and analyses will be included in the final version.

---

> ### Author Response · Authors · 2024-11-23
> **Response to Reviewer zA3J (2/3)**
>
> **Questions:**
>
> **Q1: The value of short clips.**
>
> We employed a *Clip Extraction* strategy to ensure scene consistency within each clip, resulting in the inclusion of clips shorter than 5 seconds in our dataset.
>
> These short-duration videos hold value in at least two areas:
>
> 1) **Early-stage training in T2V Generation**: Short-duration videos are particularly useful during the early stages of T2V training. Many T2V models, such as Open-sora-plan v1.2 [1] and pyramidal flow matching [2], utilize short clips in the pre-training phase to reduce complexity, gradually increasing video duration and resolution in later stages. Thus, the short clips in our dataset provide a good starting point for dynamic or multi-stage training in video generation.
>
> 2) **Applications in video restoration**: Short-duration videos are also valuable for other video generation tasks, such as video restoration. For example, video super-resolution methods [3, 4] target sequences of 8 and 6 consecutive frames respectively, while video frame interpolation techniques [5, 6] focus on 32 and 14 consecutive frames respectively.
>
> [1] PKU-Yuan Lab and Tuzhan AI etc. Open-sora-plan, apr 2024. URL https://doi.org/10.
> 5281/zenodo.10948109.
>
> [2] Jin Y, Sun Z, Li N, et al. Pyramidal flow matching for efficient video generative modeling[J]. arXiv preprint arXiv:2410.05954, 2024.
>
> [3] Zhou S, Yang P, Wang J, et al. Upscale-A-Video: Temporal-Consistent Diffusion Model for Real-World Video Super-Resolution[C]//Proceedings of the IEEE/CVF Conference on Computer Vision and Pattern Recognition. 2024: 2535-2545.
>
> [4] Chen Z, Long F, Qiu Z, et al. Learning Spatial Adaptation and Temporal Coherence in Diffusion Models for Video Super-Resolution[C]//Proceedings of the IEEE/CVF Conference on Computer Vision and Pattern Recognition. 2024: 9232-9241.
>
> [5] Jain S, Watson D, Tabellion E, et al. Video interpolation with diffusion models[C]//Proceedings of the IEEE/CVF Conference on Computer Vision and Pattern Recognition. 2024: 7341-7351.
>
> [6] Wang W, Wang Q, Zheng K, et al. Framer: Interactive Frame Interpolation[J]. arXiv preprint arXiv:2410.18978, 2024.
>
> **Q2: Self-Attention Module.**
>
> As you anticipated, the self-attention operation is conducted along spatial and within the same frame. We will clarify this further in the revised version.
>
> Training a 3D full-attention architecture, as used in Open-sora-plan v1.2, requires immense GPU memory, computational resources, and time (generating a single video on an A100 GPU takes 1.5 hours), as highlighted in its report: "the substantial computational cost made it unsustainable".
>
> In contrast, our OpenVid-1M dataset is designed for high-resolution video generation. Employing 3D full attention in this context would lead to overwhelming GPU memory and computational demands, rendering model training nearly impossible. However, as noted in the Open-sora-plan v1.2 report, 3D full attention excels at capturing joint spatial-temporal features. Exploring methods to extend 3D full attention to high-resolution video generation is an important direction and will be a focus of our future research.

---

> ### Author Response · Authors · 2024-11-23
> **Response to Reviewer zA3J (3/3)**
>
> **Questions:**
>
> **Q3. SOTA list update.**
>
> Following your advice, we compared with the SOTA CogvideoX-5B and Open-sora-plan v1.2 in the table below:
>
> |  | Resolution | Training Data | VQA$_A$$\uparrow$ | VQA$_T$$\uparrow$ | Blip Bleu$\uparrow$ | SD Score$\uparrow$ | Clip Temp Score$\uparrow$ | Warping Error$\downarrow$ |
> |:------------:|:------:|:------:|:------:|:------:|:------:|:------:|:------:|:------:|
> | OpenSoraPlan-V1.2 | 640×480 | Self collected-7.1M | 23.25 | 65.86 | 19.93 | 69.21 | 99.97 | 0.001 |
> | CogvideoX-5B | 720x480 | Self collected-35M | 35.12 | 76.86 | 24.21 | 68.91 | 99.79 | 0.0006 |
> | Ours | 1024×1024 | OpenVid-1M | 73.46 | 68.58 | 23.45 | 68.04 | 99.87 | 0.0052 |
>
> The table shows that our model achieves superior aesthetic performance and comparable clarity to Open-sora-plan v1.2 and CogVideoX-5B while using less training data and parameters, demonstrating the strength of our dataset in ensuring high quality. These results will be included in the final version.
>
> **Q4. Fair comparison with Panda-50M**
>
> Thank you for the valuable suggestions. In the table below, we select 1080P videos from Panda-50M to create Panda-50M-HD and compare the model performance between OpenVid-1M and Panda-50M-HD.
>
> | Resolution | Training Data | VQA$_A$$\uparrow$ | VQA$_T$$\uparrow$ | Blip Bleu$\uparrow$ | SD Score$\uparrow$ | Clip Temp Score$\uparrow$ | Warping Error$\downarrow$ |
> |:------------:|:------:|:------:|:------:|:------:|:------:|:------:|:------:|
> | 1024×1024 | Panda-50M (4× Super-resolution) | 63.25 | 53.21 | **23.60** | 67.44 | 99.57 | 0.0163 |
> | 1024×1024 | Panda-50M-HD | 13.48 | 42.89 | 21.78 | **68.43** | 99.84 | 0.0136 |
> | 1024×1024 | OpenVidHD-0.4M (Ours) | **73.46** |  **68.58** | 23.45 | 68.04 | **99.87** | **0.0052** |
>
> From the table, we observe that the model trained on our OpenVid-1M demonstrates superior performance, while the model trained on Panda-50M-HD performs poorly. This discrepancy may be attributed to the *low-quality videos* in Panda-50M-HD (e.g., low aesthetics and clarity, nearly static scenes, and frequent flickering), a problem our data processing pipeline effectively avoids.

---

> ### Comment · Reviewer_zA3J · 2024-11-25
>
> Thanks for the detailed response.  It is nice  that the authors have tried their best to address most of my technique concern.  On the other hand, there are some intrinsic drawbacks ,  including the videos are not first-hand data and the novelty of the model architecture designing,   which can not be addressed in this work.  Hence,  I keep my rating as 6,  and look forward to more solid sequel in the future (if this work is accepted).

---

> > ### Author Response · Authors · 2024-11-25
> > **Response to Reviewer zA3J**
> >
> > Thank you for your response! We’re delighted to hear that our reply addressed most of your technique concerns and appreciate your recognition of our efforts. We truly appreciate your thorough and valuable feedback, which has greatly helped improve our work.
> >
> > **The value of OpenVid-1M.**
> >
> > Previous datasets contain a significant amount of *noisy* data, which severely impacts the performance of models trained on them. As demonstrated in Table 7(*Paper*) and the additional Panda-50M-HD experiments in **Q4 of Questions**, this issue is evident. Therefore, constructing the high-quality OpenVid-1M dataset using our novel pipeline is both meaningful and essential to prevent models from learning *useless or harmful* knowledge.
> >
> > Furthermore, our dataset is also valuable in other research tasks, such as video super-resolution or frame interpolation, which currently lack large-scale, high-quality video datasets like OpenVid-1M, as discussed in Section 6.2 (*Paper*).
> >
> > Last but not least, we will continue to collect higher-resolution, longer-duration video data to further enhance OpenVid-1M, similar to the efforts we have made in improving captions.
> >
> > **The novelty of MVDiT.**
> >
> > On the one hand, we designed three effective modules to cope with video data and empirically verified their effectiveness. Each of these modules has clear motivations and is complementary to the others.
> >
> > On the other hand, we specifically build our model upon MMDiT and extend it to videos, following a commonly adopted research approach, similar to how Video LDM [CVPR 2023] extended LDMs for video synthesis, TubeDETR [CVPR 2022] and TransVOD [TPAMI 2022] extended DETR to video object detection. These works are highly recognized for their contributions to addressing specific challenges in their respective fields.

---

### Meta-Review · Area_Chair_LRjQ · 2024-12-16

**Metareview:**

This paper proposes a high-quality large-scale text-to-video (T2V) dataset called OpenVid-1M, alongside a T2V  model MVDiT based on MMDiT. Reviewers acknowledged the contributions of the dataset, the model design, good results and the open source. The initial major concerns include no new videos added to existing data, and  the lack of ablations on video captioning models and model components. The rebuttal addressed most of these concerns, and the four reviewers unanimously provided positive scores (8866). Therefore, the AC recommends acceptance.

**Additional Comments On Reviewer Discussion:**

Reviewers acknowledged the contributions of the dataset, the model design, good results and the open source. The initial major concerns include no new videos added to existing data, and the lack of ablations on video captioning models and model components. The rebuttal addressed most of these concerns, and the four reviewers unanimously provided positive scores (8866).

---

### Decision · Program_Chairs · 2025-01-22

Accept (Poster)